# Parvovirus minute virus of mice interacts with sites of cellular DNA damage to establish and amplify its lytic infection

Kinjal Majumder[1]*, Juexin Wang[2,3], Maria Boftsi[4], Matthew S Fuller[5], Jordan E Rede[1], Trupti Joshi[2,3,6,7], David J Pintel[1]*

[1]Department of Molecular Microbiology and Immunology, Christopher S. Bond Life Sciences Center, Columbia, United States; [2]Department of Electrical Engineering and Computer Science, Christopher S. Bond Life Sciences Center, Columbia, United States; [3]Christopher S. Bond Life Sciences Center, Columbia, United States; [4]Pathobiology Area Graduate Program, Christopher S. Bond Life Sciences Center, Columbia, United States; [5]Ultragenyx Pharmaceutical, Christopher S. Bond Life Sciences Center, Columbia, United States; [6]Department of Health Management and Informatics, School of Medicine, University of Missouri-Columbia, Columbia, United States; [7]MU Informatics Institute, University of Missouri-Columbia, Columbia, United States

*For correspondence:
km3k5@health.missouri.edu (KM);
pinteld@missouri.edu (DJP)

**Abstract** We have developed a generally adaptable, novel high-throughput Viral Chromosome Conformation Capture assay (V3C-seq) for use in *trans* that allows genome-wide identification of the direct interactions of a lytic virus genome with distinct regions of the cellular chromosome. Upon infection, we found that the parvovirus Minute Virus of Mice (MVM) genome initially associated with sites of cellular DNA damage that in mock-infected cells also exhibited DNA damage as cells progressed through S-phase. As infection proceeded, new DNA damage sites were induced, and virus subsequently also associated with these. Sites of association identified biochemically were confirmed microscopically and MVM could be targeted specifically to artificially induced sites of DNA damage. Thus, MVM established replication at cellular DNA damage sites, which provide replication and expression machinery, and as cellular DNA damage accrued, virus spread additionally to newly damaged sites to amplify infection. MVM-associated sites overlap significantly with previously identified topologically-associated domains (TADs).
DOI: https://doi.org/10.7554/eLife.37750.001

## Introduction

DNA viruses that replicate in the nucleus depend on host cellular functions for transcriptional and replication machinery to express and amplify their genomes. Accessing these functions is critical to productive infection, yet successful establishment of replication must also overcome cellular antiviral activity, which for larger DNA viruses includes innate immune responses, epigenetic silencing, the cellular DNA-damage response (DDR), and antiviral activity found associated with PML bodies (*Weitzman et al., 2010*). Replication of many DNA viruses takes place in distinct micro-nuclear compartments termed replication centers that are rich in factors viruses must interact with - either positively or negatively - to productively replicate (*Schmid et al., 2014*). It is not fully clear, however, how nuclear-replicating viruses initiate replication centers in order to optimize access to factors and functions they need to either utilize or inactivate.

**eLife digest** Viruses are small infectious particles that can only reproduce with the help of a host. Once they are inside their victim, they hijack the cells' genetic material and reprogram it to become a virus factory that produces more virus particles. Parvoviruses, for example, are among the simplest of viruses and need all resources a cell has to offer to successfully replicate.

This process often takes place at so-called replication centers that contain these necessary factors. It was previously thought that parvoviruses set up such centers randomly, and gather the required molecules such as proteins to these sites. However, it was not well understood how they do this.

Now, Majumder et al. have developed a new method that enabled them to study in detail how parvoviruses gain access to the resources of the cell they need to initiate and amplify replication. The results show that parvoviruses set up their replication centers at sites on the host DNA that are already rich in proteins needed to repair and then replicate damaged DNA. Some of these sites already exist in the cell's genetic material as a consequence of naturally occurring processes, but others are created during infection by the virus. These findings may have important implications for how other viruses may establish their replication.

Viruses, including parvoviruses, are important pathogens. Like many microbes, viruses can be beneficial for our health and environment. Others, however, can be harmful. A clearer understanding of how viruses establish and amplify an infection may provide new treatment opportunities.

DOI: https://doi.org/10.7554/eLife.37750.002

Parvoviruses are small non-enveloped icosahedral viruses that are important pathogens in many animal species including humans. Minute Virus of Mice (MVM) is an autonomously replicating parvovirus that is lytic in murine cells and transformed human cells (*Cotmore and Tattersall, 2014*). The viral genome is approximately 5 kb and possesses inverted terminal repeats at each end that serve as origins of replication (*Cotmore and Tattersall, 2014*). MVM encodes two non-structural proteins: the larger non-structural phosphoprotein NS1 performs a number of functions required for viral replication, while NS2 plays important, currently undefined, roles during infection of the normal murine host (*Cotmore and Tattersall, 2014*).

Parvoviruses are the only known viruses of vertebrates that contain single-stranded linear DNA genomes, and thus, they present novel replicative DNA structures to cells during infection. They depend heavily on cellular functions for replication, and unlike the DNA tumor viruses, do not drive quiescent cells into S-phase. However, following S-phase entry, cellular DNA polymerase δ converts the single stranded viral DNA genome into a double stranded molecule that serves as a template for transcription of the viral genes (*Cotmore and Tattersall, 2013*). As MVM infection progresses through S-phase, it induces substantial cellular DNA damage and evokes a robust, ATM-dependent DNA damage response (DDR, [*Adeyemi et al., 2010*]). Infection is characterized by a pre-mitotic cell cycle arrest that is both p21 and CHK1 independent (*Adeyemi and Pintel, 2012*; *2014*). During this block virus replication proceeds for many hours, and ATM inhibitors reduce ongoing viral replication (*Adeyemi et al., 2010*).

Parvoviruses establish replication factories in the nucleus (termed Autonomous Parvovirus-Associated Replication, or APAR, bodies) where active transcription of viral genes and viral replication takes place (*Bashir et al., 2001*). Low resolution confocal microscopy originally showed that DDR sensor and response proteins, cell cycle regulators, DNA polymerases and RNA polymerase II accumulate in MVM APAR bodies where they co-localize with replicating viral DNA and NS1 (*Adeyemi et al., 2010*; *Bashir et al., 2001*; *Ruiz et al., 2011*). More recently, higher magnification microscopic studies revealed that phosphorylated histone H2A variant γ-H2AX, as well as other DDR proteins ([*Ruiz et al., 2011*] see *Figure 1*, below), seemed to reside adjacent to, rather than within, early APAR bodies. This gave rise to the suggestion that these DDR factors may reside on cellular DNA in the vicinity of viral replication centers (*Ruiz et al., 2011*). The induction of cellular DNA damage leads to almost instantaneous recruitment of DDR factors to the break site which coordinate complex signaling cascades that recruit DNA damage sensors, repair mediators and effectors (*Hashiguchi et al., 2007*; *Polo and Jackson, 2011*). Endogenous sites of damage are also often

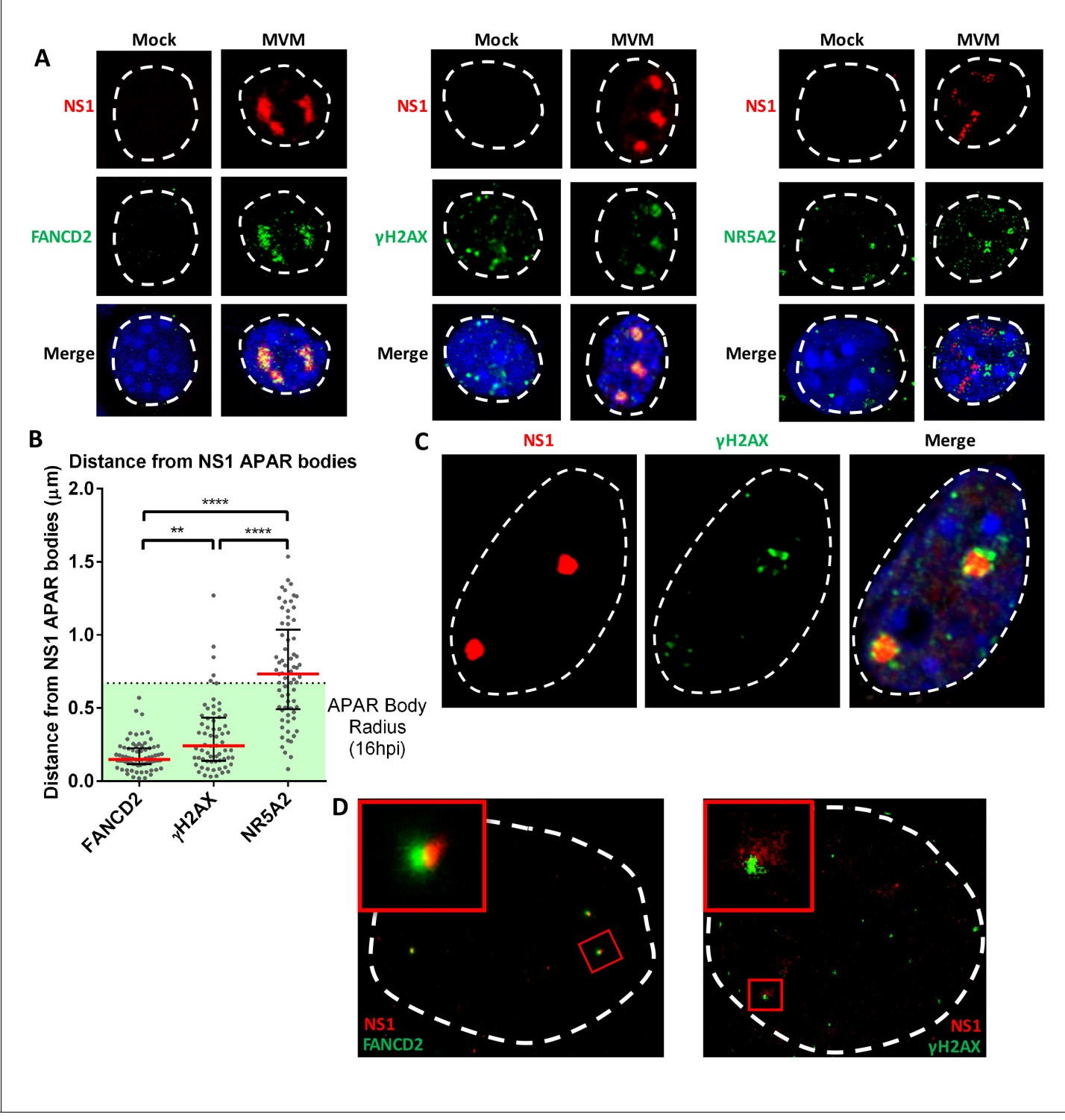

**Figure 1.** The replicating MVM genome associates with cellular sites undergoing DNA damage. (**A**) Representative confocal images of Mock versus MVM infected murine A9 cells at 16 hpi, probing MVM-NS1 (*red*) and the DDR factors FANCD2 and γ-H2AX, and the irrelevant transcription factor NR5A2 (*green*), quantified in (**B**). Blue corresponds to DAPI staining. Nuclear border is indicated by dashed white line. (**B**) The distances between NS1 and indicated DDR proteins were calculated from deconvolved confocal z-stacks using Huygens and ImageJ programs (described in Materials and methods), and the non-associated transcription factor NR5A2 was used as a negative control. Results are represented as grey scatterplots from three independent infections, with the median value of the dataset depicted by a red line. Black error bars represent the interquartile range of the dataset. The radius of APAR bodies were calculated by measuring the diameter of APAR bodies from multiple fields from three independent infections at 16 hpi

*Figure 1 continued on next page*

Figure 1 continued

imaged using confocal microscopy and deconvolved using Huygens software. The radius of APAR body was calculated by dividing the median diameter by 2, and is represented as a dashed horizontal line. Significant differences are denoted as *p<0.05, **p<0.005 and ****p<0.0005 (one-way ANOVA, multiple comparisons). (C) Representative image of an APAR body at 16 hpi imaged using a super-resolution Airyscan imaging platform, where NS1 (red) stains the APAR body and γ-H2AX (green) stains for DNA damage. DAPI stains the nuclear border, demarcated by a white dashed line. (D) Super-resolution GSD-STORM imaging of MVM-APAR bodies at 16 hpi and cellular DDR markers, including FANCD2 (left) and γ-H2AX (right). The nuclear borders are demarcated by a dashed white line and were identified by brightfield imaging (not shown). The inset shows magnifications of the APAR bodies (red) and the respective DDR protein (green).

DOI: https://doi.org/10.7554/eLife.37750.003

associated with interference between replication and transcription polymerases (*Durkin and Glover, 2007*). DNA breaks, therefore, serve as cellular depots of DDR proteins, and factors involved in DNA replication and expression (*Hashiguchi et al., 2007*; *Polo and Jackson, 2011*). While it is plausible that during infection damaged cellular DNA is relocated to sites of MVM replication, it seemed possible, alternatively, that the virus initially established its replication centers at damaged cellular sites where these DDR, replication, and expression factors were already present.

It has been suggested that a number of viruses, including hepatitis B virus (HBV) and human papillomavirus (HPV), associate with sites of DNA damage, including early-replicating or common, fragile sites (ERFs, and CFSs, respectively) at early times during their infections (*Durkin and Glover, 2007*; *Jang et al., 2014*; *Tubbs and Nussenzweig, 2017*). The HPV genome was shown to be tethered to CFSs by the chromatin modifier BRD4, which facilitates subsequent integration into the host genome utilizing the cellular DDR machinery (*Feitelson and Lee, 2007*; *Jang et al., 2014*). These studies utilized chromatin immunoprecipitation (ChIP) assays of the HPV E2 protein to identify sites of HPV localization to cellular CFSs, which were then validated by 3D-FISH assays of the viral and cellular DNA (*Jang et al., 2014*). Consistent with these findings, crosslinked-ChIP assays have recently demonstrated that FANCD2, required for maintaining fragile site stability and coordinating their replication, also associates with HPV genomes at replication centers (*Madireddy et al., 2016*; *Spriggs and Laimins, 2017*). However, an unbiased way to map the interaction between the viral and cellular genomes directly has been largely unavailable until recently.

The development of Chromosome Conformation Capture (3C) technologies has enabled detailed analysis of *cis*-interactions between separated regions of the genome, as well as aspects of chromatin packaging (*Dixon et al., 2016*). When combined with high-throughput sequencing, these techniques (termed 4C, 5C, and Hi-C assays) have become valuable tools for studying the details of nuclear configuration (*Denker and de Laat, 2016*). Specifically, 4C and Hi-C assays provide genome-wide interaction data through unbiased deep sequencing. Hi-C assays are designed to provide information on all nuclear interactions whereas 4C assays enable higher resolving power and a deeper interrogation of genomic associations by utilizing inverse PCRs to amplify only the 'bait' (*Lajoie et al., 2015*). In this study, we report the novel adaptation of high-throughput circular chromosome conformation capture assay (4C) for use in trans which we term V3C (Viral Chromosome Conformation Capture). This assay, which should be generally adaptable, has allowed us to characterize, on a genome-wide scale, the direct association of the linear MVM genome with discreet regions of the cellular genome. These sites, termed Virus Association Domains (VADs), correlated initially with sites of cellular DNA damage that in mock-infected cells also exhibited damage as cells progressed through S-phase. As infection progressed these sites expanded, and additional sites of DNA damage were induced. MVM subsequently associated with the newly induced sites as infection amplified. Sites of association identified biochemically were confirmed microscopically, and in addition, MVM could be targeted specifically to sites of DNA damage artificially engineered into the cellular chromosome. Development of the V3C assay has allowed us to suggest the following model. Soon after nuclear entry MVM homes first to sites of pre-existing endogenous DNA damage to initiate infection at sites that provide cellular factors necessary for its replication. Subsequently, as cellular DNA damage accrues, virus spreads additionally to these sites of damage to amplify infection. The V3C-seq assay should be useful for characterizing the interaction of many DNA viruses that associate with the cellular genome, and provide a useful tool to characterize the molecular events leading to the initiation of infection.

## Results

### The replicating MVM genome localized adjacent to regions of the cellular genome undergoing a DDR

As MVM infection progresses through S-phase, it induces cellular DNA damage and evokes a robust, ATM-dependent DNA damage response (DDR) characterized by a pre-mitotic cell cycle arrest that is both p21 and CHK1 independent (*Adeyemi and Pintel, 2012*, *2014*). During this block, virus replication proceeds for many hours, and ATM inhibitors reduce ongoing viral replication (*Adeyemi et al., 2010*). Standard confocal microscopy demonstrated previously that numerous cell cycle and DDR effector proteins, RNA polymerase II, as well as DNA polymerase-α and δ are found associated with MVM replication centers called APAR bodies (*Adeyemi et al., 2010*; *Bashir et al., 2000*; *Kollek et al., 1982*; *Ruiz et al., 2011*). Similar examples of such confocal images of DDR proteins associated with APAR bodies, but not the irrelevant transcription factor NR5A2 (*Duggavathi et al., 2008*), processed with deconvolution, can be seen in *Figure 1A*. Comparison of the median three-dimensional distance between FANCD2 and γ-H2AX - which localize to stalled replication forks where they facilitate DNA repair (*Kim et al., 2018*; *Lossaint et al., 2013*; *Madireddy et al., 2016*), and the center of APAR bodies - identified by MVM NS1 staining, indicated that at 16 hr post infection (hpi) and release (representing approximately 8–10 hr into S-phase in our para-synchronization protocol), they localized closely with MVM replication centers (*Figure 1B*). This was in contrast to the irrelevant transcription factor NR5A2 (*Duggavathi et al., 2008*) which exhibited a diffuse localization (*Figure 1B*). However, confocal super resolution imaging (Airyscan) demonstrated that γ-H2AX seemingly localized to the periphery of APAR bodies (*Figure 1C*). Super-resolution imaging using the GSD-STORM platform also demonstrated that γ-H2AX, and FANCD2 (*Figure 1D*), localized to the periphery of APAR bodies (*Figure 1D*). γ-H2AX characteristically amplifies on damaged DNA as megabase (Mb)-sized platforms, making the possibility of marking the 5 kb MVM genome with phosphorylated histone H2AX less likely (*Rogakou et al., 1999*). These results suggested that, MVM replication centers may localize to and expand adjacent to sites of cellular chromatin undergoing DNA damage, where replication, expression and DDR factors reside.

### The MVM genome associated directly with discrete sites on the cellular genome

To characterize the association of the MVM genome with the cellular genome during lytic infection more directly we have developed a high-throughput chromosome conformation capture (3C) assay for use in trans. 3C assays have been typically used to identify long-range interactions between regions of a single chromosome (*Dekker et al., 2013*). Our analysis, which we term V3C-seq (Viral Chromosome Conformation Capture Sequencing) allowed us to identify direct interactions between the linear MVMp genome and the cellular chromosome in an infected cell population on a genome-wide scale in an unbiased manner. V3C-seq assays utilize formaldehyde-mediated crosslinking to first 'freeze' the locations of the viral and cellular genomes at various points during infection. Samples are then digested and ligated under conditions that favor intramolecular interaction, and the resultant novel virus-cell DNA fragments are subjected to high-throughput sequencing (*Figure 2A*). The assay provides a precise genomic map of the sites with which viral DNA interacts, and the frequency with which unique individual linked fragments are generated provides quantification of these interactions.

V3C-seq was performed in parasynchronized mouse A9 fibroblasts, the traditional host for MVM, at various times post-infection. A typical time-course of MVM infection is shown in *Figure 2—figure supplement 1A*. Assays utilized a viral viewpoint at a HindIII site at nucleotide 2651 and NlaIII site at 1899 in the MVMp genome, thereby capturing the interaction of the MVMp fragment containing both the viral P4 and P38 promoters upstream of the HindIII site. Clustering algorithms and visualization of interaction sites on the UCSC Genome Browser (*Kent et al., 2002*; *Ramírez et al., 2016*) revealed that by 12 hr post-infection and release (representing approximately 4–6 hr into S-phase) MVM genomes associated with discrete regions on most cellular chromosomes [*Figure 2—figure supplements 1B* and *2*, (*Kent et al., 2002*)]. These cellular sites served as initial amplification points for MVM, and upon progression to 16 hpi, the virus both expanded at these regions and associated

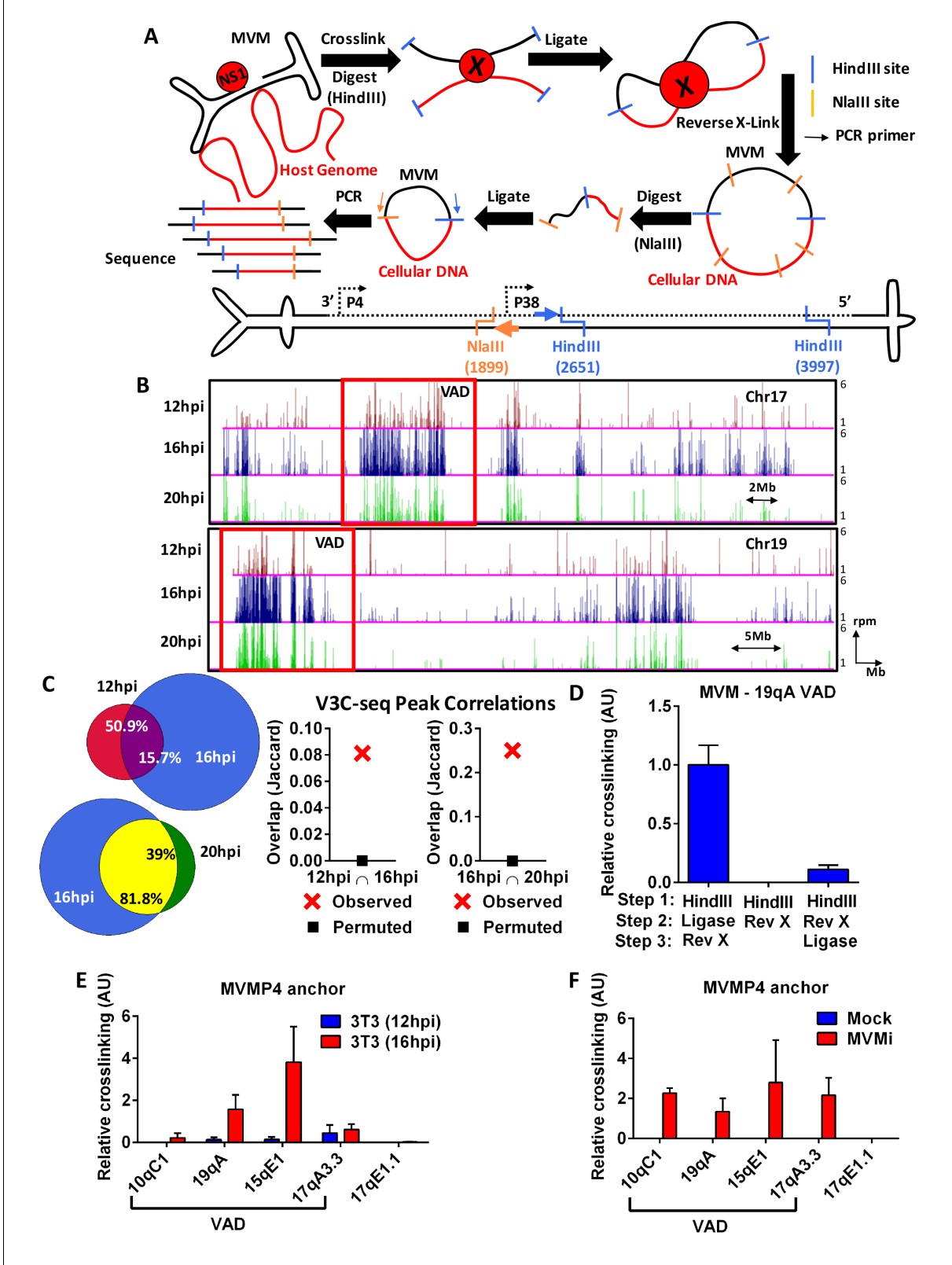

**Figure 2.** The MVM genome associates with distinct sites on the cellular genome. (**A**) *Top* Schematic of the V3C-seq assay showing how MVM- host cell genomic proximity is frozen by crosslinking, followed by digesting (with HindIII) and intramolecularly ligating to generate novel MVVM-host cell DNA hybrids. This DNA library is subjected to a second round of digestion with a frequently-digesting 4 base-pair endonuclease (NlaIII), before circularizing and generating a sequencing library of all hybrid fragments that associate with the MVM genome. *Bottom* Detailed schematic of the duplex form of

*Figure 2 continued*

MVMp genome containing the primary restriction enzyme site (HindIII) with its associated inverse PCR primer (blue arrow), and the secondary restriction enzyme site (NlaIII) with its associated inverse PCR primer (orange arrow) utilized for circularization. The single stranded version of the genome is depicted in solid black line and complementary strand in dotted black line. (B) Associations of the MVM genome with sites on the cellular DNA mapped using V3C-seq assays are presented. Representative examples of murine chromosome 17 (*top*) and chromosome 19 (*bottom*) are shown for three timepoints post-infection: 12 hpi (*red*), 16 hpi (*blue*) and 20 hpi (*green*). The data represents an average of at least 2 independent experiments with y-axis values reflecting the reads per million (rpm) sequence reads averaged over 5 contiguous fragments (as described in Materials and methods). The y-axis scale is from 1 to 6 rpm, whereas the x-axis is 95 Mb for chromosome 17 (top) and 61 Mb for chromosome 19 (bottom). Large genomic regions that associate with MVM, termed Virus Associated Domains (VADs), are shown by red boxes, but this indication is not meant to restrict this nomenclature to regions of this size. (C) Genomic regions spanned by V3C-seq peaks greater than 5 total reads were selected, and the common regions between different timepoints were intersected using BEDTools (see Materials and methods). In the Venn diagrams on the left panel, the red set represents genomic regions that were covered by 12 hpi, blue represents 16 hpi and green represents 20 hpi. The regions common to 12 and 16 hpi are shown in purple (top), while 16 and 20 hpi are in yellow (bottom). The percent of genomic regions that are common are depicted in the intersected set. Statistical significance of the overlap was computed using Jaccard analysis on BEDtools (right panels, red crosses) between the top ten thousand V3C-seq peaks at 12, 16 and 20 hpi timepoints, with control comparisons permuted by determining the extent of overlap with a randomly generated peak file containing domains of equivalent V3C-seq peaks (represented by black squares). (D) 3C-qPCR was performed on synchronized murine A9 cells infected with MVM with an MOI of 5 for 16 hr, as described in Materials and methods, and then analyzed with the viewpoint on the MVM genome. The association of MVM with a HindIII site in the Chr19 VAD (at position 19qA) was quantified relative to nearest neighbor interactions of contiguous HindIII fragments on the *Ercc3* locus. 3C-qPCR analysis was performed in (E), parasynchronized NIH-3T3 cells infected for 12 and 16 hr with MVMp, and (F), EL4 cells with MVMi, assayed from the MVM viewpoint. Association was tested with four VADs (10qC1, 19qA, 15qE1 and 17qA3.3) and a negative control site on Chromosome 17 (17qE1.1). Data is presented as mean $\pm$ SEM of three independent experiments.

DOI: https://doi.org/10.7554/eLife.37750.004

The following figure supplements are available for figure 2:

**Figure supplement 1.** MVM replication during viral infection and correlation of V3C-seq interaction sites.
DOI: https://doi.org/10.7554/eLife.37750.005

**Figure supplement 2.** Genome browser snapshots of MVM interaction sites on all chromosomes in the mouse genome.
DOI: https://doi.org/10.7554/eLife.37750.006

with new sites (*Figure 2B*, *Figure 2—figure supplement 2*). We term these sites of association Virus Associated Domains, or VADs. The clusters of interacting sites on each chromosome ranged in density and in size (*Figure 2—figure supplement 2*), from approximately 1–2 Mb to larger 5–15 Mb-size domains (chromosomes 17 and 19 are shown in *Figure 2B*). While most chromosomes contained multiple small VADS, many contained 1–3 larger VADs of 5–15 Mb size. Chromosomes 1, 13, 18, X and Y had fewer discernible interaction sites. The larger VADs in *Figure 2B* are boxed for comparison purposes but is not meant to restrict the designation of VADs to a particular size.

Comparison of MVM interaction sites across the entire mouse genome showed that approximately 51% of VADs identified at 12 hpi were retained at 16 hpi, indicating that approximately 84% of VADs identified at 16 hpi were newly generated (*Figure 2C*). Approximately 39% of MVM interaction sites detected at 16 hpi were retained at 20 hpi, while only approximately 18% of the interaction sites identified at 20 hpi were newly generated (*Figure 2C*). Clustering analysis of MVM interaction sites in multiple replicates over the time-course of infection showed that they were reproducible across replicates (*Figure 2—figure supplement 1B*). Importantly, MVM interaction sites clustered together at 16 and 20 hpi in a characteristic manner that was distinct from early (12 hpi), and late (24 hpi) infection. These results suggested that interactions of MVM with the cellular chromosome increased as infection progressed. By 24 hpi, MVM interaction with the host chromosome was extensive (*Figure 2—figure supplement 2*). This latter observation indicated that MVM interaction with the cellular genome at VADs during early stages of infection was not an artifact of preferential sequencing at the VAD sites, and is consistent with previously published profiling of APAR bodies by microscopy (*Ruiz et al., 2011*). Additionally, at these late times, in the presence of saturating amount of MVM DNA, we observed interactions of the viral genome with sites throughout the mouse genome. This further suggested that VADs identified earlier during infection featured properties that enabled viral recruitment and replication. It is noteworthy, however, that the cellular genome undergoes substantial DNA damage by late stages of infection (*Figure 3—figure supplement 2A* and as described below), likely precluding detection of some interactions effectively by V3C-seq. In addition, viral packaging would be expected to reduce available viral genomes by

approximately 20 hpi (*Cotmore and Tattersall, 2014*), which would also be predicted to contribute to the decreased interaction seen at 20 hpi in *Figure 2B*.

We chose to validate the V3C assay by confirming the association of MVM with one of these VADs, murine 19qA, using a focused Taqman-based assay, in which the frequency of novel ligation junctions were determined by quantitative PCR (qPCR). MVM association with the VAD at 19qA was readily detectable in our standard protocol using Taqman probes complementary to the MVM genome (forward) and the cellular genomic site (reverse) (*Figure 2D*). This association was substantially diminished, either in the absence of intramolecular ligation, or when cross-links were reversed prior to intramolecular ligation (*Figure 2D*). These experiments defined the lower limit of background levels generated by our V3C assay, and suggested that MVM associations with cellular VADs were specific, and mediated by DNA-DNA and/or DNA-protein intramolecular crosslinks.

Focused 3C-qPCR in parasynchronized NIH-3T3 fibroblasts infected with MVMp (*Figure 2E*), and EL4 lymphocyte cells infected with the lymphotrophic variant MVMi (*Figure 2F*) both showed association with a subset of the VADs identified in A9 cells (*Figure 2B*, *Figure 2—figure supplement 2*), which suggested a common mechanism may exist for the establishment of MVM replication sites in these mouse cell lines. Differential rates of MVM replication and cell cycle kinetics in A9 compared to NIH-3T3 and EL4 cells precluded performance of 3C-qPCR assays simultaneously in these systems. Together, we interpret our results as establishing that V3C-seq is a valid means to map the interaction of a lytic linear DNA virus with specific sites on the host cell genome.

## MVM infection induced distinct sites of cellular DNA damage as demonstrated by ChIP-seq for γ-H2AX

As described above, super-resolution microscopy suggested that MVM replication centers seemingly associated adjacent to genomic sites containing factors involved in replication, expression, and the DDR. ERFs have such characteristics in uninfected cells, and as mentioned, MVM continues to induce DNA damage as infection proceeds (*Adeyemi et al., 2010*; *Barlow et al., 2013*). Therefore, as we identified sites of viral interaction with the cellular genome, we looked for association with sites of cellular DNA damage.

To identify sites of cellular DNA damage, we initially performed chromatin immunoprecipitation coupled with high-throughput sequencing (ChIP-seq, [*Landt et al., 2012*]) for γ-H2AX in parasynchronized A9 cells, either mock infected, infected with MVM, or mock infected and treated with hydroxyurea (HU). Results for algorithm-called peaks for chromosomes 17 and 19 are shown in *Figure 3A*, and for the complete murine genome is shown in *Figure 3—figure supplement 1*. Mock infected A9 cells (taken 12 hr post release, hpr) showed a significant number of sites of damage, as identified by γ-H2AX, as they passed into S-phase. These were likely ERFs, which accrue damage during replication (*Figure 3A*). Whole-genome peak analysis of all γ-H2AX bound regions revealed that approximately 55% of the sites identified in mock infected cells overlapped with those identified at 16 hr post-infection (hpi); approximately 55% of sites identified at 16 hpi were newly generated (*Figure 3B*, peak calling using EPIC and intersection using BEDtools, (*Quinlan and Hall, 2010*), as described in Materials and methods). By 16 hpi, sites of damage concentrated in distribution, and expanded in number (*Figure 3A*). At this point in infection MVM had begun to induce additional sites of DNA damage, as evident both by ChIP-seq and increased tail moments in Comet assays (*Figure 3—figure supplement 2A*). The majority of γ-H2AX-containing sites at 20 hpi were newly generated, coinciding with only 10% of the sites identified at 16 hpi, indicating the widespread induction of DNA damage by this point of infection. This can be seen more clearly in the magnified view of chromosomes 17 and 19 shown in *Figure 3—figure supplement 2C*. Interestingly, the γ-H2AX ChIP-seq regions identified at 16 hpi correlated well with γ-H2AX ChIP-seq performed following 12 hr treatment with HU (*Figures 3A* and 16 hpi vs HU). Approximately 51% of γ-H2AX peaks detected 16 hpi were shared with those induced after 12 hr treatment with HU, and conversely, approximately 26% of the peaks identified following treatment with HU were shared at 16 hpi. In order to confirm the statistical significance of the intersection analyses, the γ-H2AX peaks at indicated time points were intersected with randomly permuted peaks across the mouse genome and visualized as a Jaccard Plot (*Figure 3B*, far right).

The MVM genome initiated infection at sites of cellular DNA damage that in mock infected cells also exhibited DNA damage as the cells cycled through S-phase, and as infection progressed, localized to additional sites of induced damage.

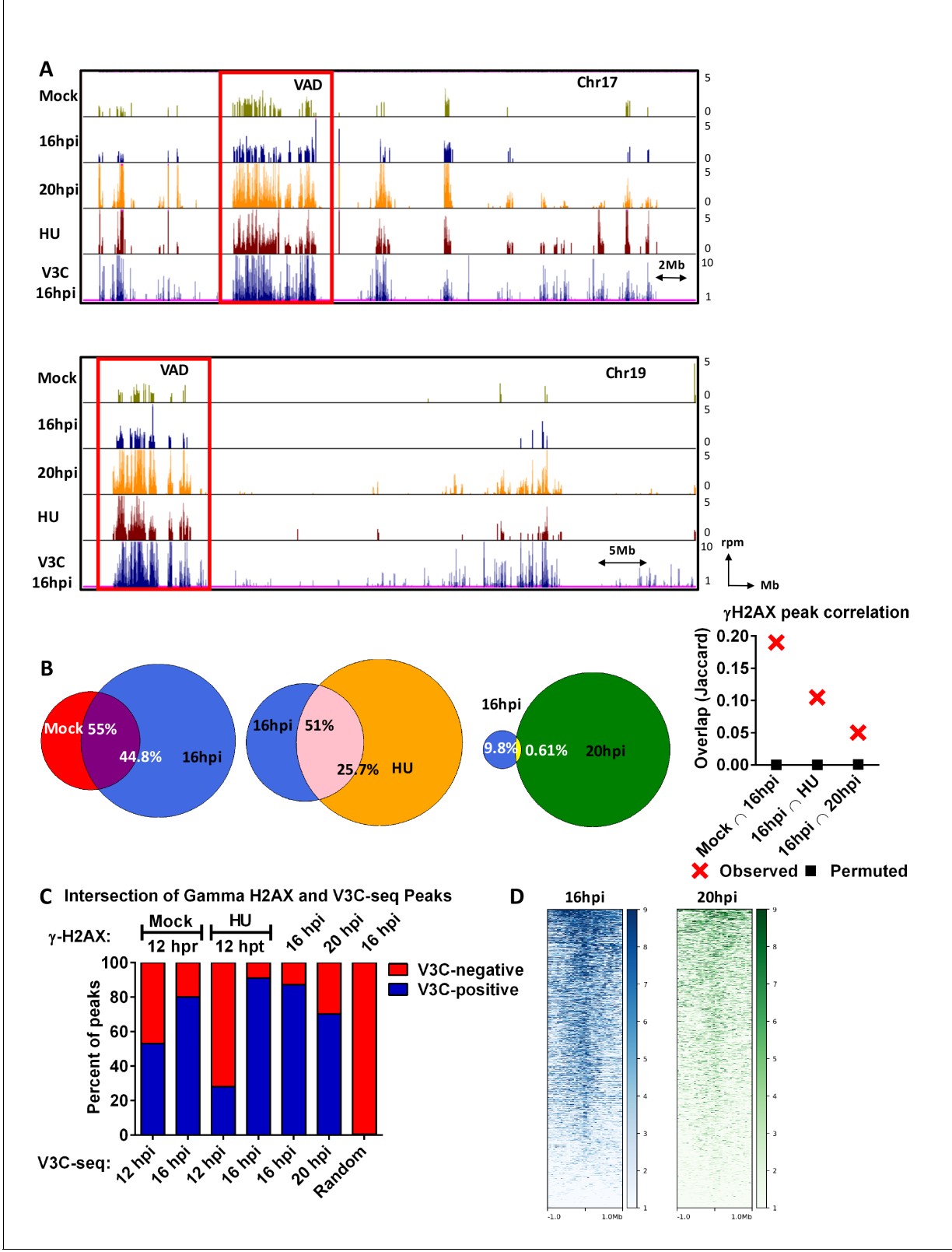

**Figure 3.** The MVM genome initiated infection at sites of cellular DNA damage that in mock infected cells also exhibited DNA damage as the cells cycled through S-phase, and as infection progressed, localized to additional sites of induced damage. (**A**) Representative quantile normalized ChIP-seq plots of γ-H2AX binding to the cellular genome on Chromosome 17 (top) and Chromosome (19). The tracks represent γ-H2AX ChIP-seq peaks in A9 cells that are mock infected at 12 hr post release (green), MVM infected at 16 hpi (blue), 20 hpi (yellow), and HU treated A9 cells 12 hr (maroon). V3C-
*Figure 3 continued on next page*

*Figure 3 continued*

seq peaks at 16 hpi are also shown at the bottom for comparison (blue). Red rectangle denotes VAD sites, and y-axis values for ChIP-seq peaks have been restricted from 0 to 5 reads per million. (B) The EPIC-called ChIP-seq peaks at different timepoints for MVM infection were analyzed for coincident γ-H2AX binding using BEDTools (*Quinlan and Hall, 2010*), and the resulting distances covered on the genome were plotted on Venn Diagrams as percentage of total coverage. Statistical significance of the overlap was determined using Jaccard analysis on BEDtools (far right, red crosses), with a control comparison permuted by determining the extent of overlap with a randomly generated peak file with domains of equivalent size as ChIP-seq peaks (represented as black squares). (C) γ-H2AX peaks from ChIP-seq experiments were intersected with VAD peaks identified in *Figure 1C* for the corresponding timepoints (Mock γ-H2AX was intersected with 12 hpi). The percentage of total regions that coincided were calculated and plotted for VAD-associated γ-H2AX peaks (designated as 'V3C-positive'), and γ-H2AX associated VAD-peaks. (D) Heatmap of MVM association with DNA damaged sites were generated using DeepTools on the Galaxy server (*Afgan et al., 2016*; *Ramírez et al., 2016*). The average V3C enrichment on 1 Megabase around γ-H2AX positive sites were determined and plotted as shown.

DOI: https://doi.org/10.7554/eLife.37750.007

The following figure supplements are available for figure 3:

**Figure supplement 1.** Genome browser snapshots of γ-H2AX occupancy over the entire mouse genome at different timepoints of MVM infection.
DOI: https://doi.org/10.7554/eLife.37750.008

**Figure supplement 2.** MVM DNA damage interactions.
DOI: https://doi.org/10.7554/eLife.37750.009

Comparisons of the ChIP-seq results with V3C-seq assays showed that MVM associated directly with sites of cellular DNA damage, as identified by the presence of γ-H2AX at the same region, in a manner that increased as infection progressed. *Figure 3A* compares MVM VADs at 16 hpi, to sites of DNA damage (as determined by γ-H2AX ChIP-seq) for chromosomes 17 and 19 as infection progressed. Large VAD regions in *Figure 3A* are boxed for comparison purposes, but are not meant to restrict overlap only to VADs of that size. Comparisons for the full mouse genome are shown in *Figure 3—figure supplement 1* and while there is significant variation, the overlap between VADs and sites positive for γ-H2AX ChIP-seq was strikingly consistent.

*Figure 3C* summarizes the genome-wide correlation at the nucleotide level of VADs and γ-H2AX ChIP-seq data presented in *Figure 3—figure supplement 1*. For the composite comparisons at 12, 16 and 20 hr post-infection, data was taken from the same experiment (comparing VADs at various time points as shown in *Figure 2B* to ChIP-seq sites at those times as shown in *Figure 3A*). At 12 hpi, MVM associated with approximately 55% of sites that in mock infected cells exhibited DNA damage upon progression into S-phase. By 16 hpi, this association rose to close to 80%.

By 16 hpi, close to 90% of γ-H2AX occupied sites overlapped with VADs (*Figure 3C*), which included γ-H2AX sites present in uninfected cells (*Figure 3A*). By the late time point of 20 hpi, approximately 70% of the γ-H2AX sites co-localized with VADs. Visualization of MVM association in the vicinity of γ-H2AX-positive sites using hierarchical clustering further revealed that MVM association with damaged sites increased from 16 hpi to 20 hpi; however, increased incidence of cellular DNA breaks after 20 hpi led to a decrease in the proportion of MVM-associated damaged sites (*Figure 3D*). Notably, approximately 25% of the VADs identified at 12 hpi, and approximately 95% of VADs identified at 16 hpi, associated with the γ-H2AX sites identified following 12 hr of treatment with HU. Randomly generated peaks showed less than 1% overlap with γ-H2AX peaks identified 16 hpi (*Figure 3C*). A magnified view of the large VADs at 19qA and 17qA/B outlined in *Figure 3A* is also provided in *Figure 3—figure supplement 2C* (left), while further magnifications of VAD regions demarcated by red rectangles in *Figure 3—figure supplement 2C* (left panel) at *Narfl*, *Vwa7*, *Ehd1* and *Slc29a2* genes are shown on the right panel.

The strong correlation of MVM interaction sites with sites that in uninfected cells exhibit DNA damage upon replication is consistent with the notion that MVM may have initially established replication at cellular fragile sites that are susceptible to DNA damage as cells cycle through S-phase, although it cannot be formally ruled out that these sites were virally-induced at the earliest times in infection. It is also important to note that VADs also correlated strongly with γ-H2AX ChIP-seq sites identified on cellular chromosomes of non-infected, HU-treated A9 cells (*Figure 3A*, red rectangles). This strongly implied, although does not prove, that the γ-H2AX identified associated with MVM during infection resides on cellular DNA. It is striking that sites of damage in uninfected cells, sites of damage induced by virus, and sites of damage induced by treatment of the DNA-damaging agent HU overlap so significantly.

Cellular sites of DNA damage also often contain BRCA1, which binds DNA and can co-localize with γ-H2AX in DNA double-strand break repair foci, although typically in a more narrow pattern (*Barlow et al., 2013*). As expected, VADs also strongly associated with sites identified by BRCA1 ChIP-seq at 16 hr post-infection (*Figure 4A*, row 2). Furthermore, VADs also overlapped with BRCA1 and γ-H2AX sites in primary mouse cells induced with replication stress agents (*Figure 4—figure supplement 1A and B*), and characterized as ERFs, in previously published studies (*Barlow et al., 2013*). As MVM can infect transformed human cells, we also performed focused V3C-qPCR at 16 hpi in parasynchronized SV40-transformed human NB324K cells. As shown in *Figure 4—figure supplement 1C*, MVM localized to the previously characterized human fragile site FRA5H, but not FRA11F. MVM also associated weakly at this time point with the prototypical human fragile site, FRA3B.

As an indirect confirmation of MVM recruitment to the cellular genome, we performed ChIP-seq for the MVM-NS1 protein, which binds covalently to the viral genome, and non-covalently to additional ACCAACCA consensus sequences throughout the MVM genome (*Christensen et al., 1995*). We reasoned that ChIP-seq assays for NS1 would confirm cellular sites associated with the viral DNA by secondary crosslinking of NS1-bound MVM DNA to cellular DNA. *Figure 4A* shows NS1 binding profiles to cellular chromosome 17 and 19 that are concordant with the VADs at 16 hpi, further validating our findings from V3C-seq assays. A genome-wide analysis of called peaks indicated that approximately 90% of the peaks identified by NS1-, BRCA1 -, and γ-H2AX ChIP-seq overlapped with VADs identified by V3C (*Figure 4B and C*), while overlap was undetectable when intersected with a randomly generated library of ChIP-seq peaks of equivalent size (*Figure 4B*). In concordance with these findings, the binding profile of NS1, BRCA1 and γ-H2AX around a VAD site at 16 hpi centered within 1 Mb of the MVM associated cellular site (*Figure 4B*). Taken together, our V3C-seq and ChIP-seq experiments are consistent with a model that upon infection, MVM first localized to cellular sites susceptible to DNA damage as cells progressed into S-phase, and as infection progressed, localized to additional sites of damage that were virally induced, to amplify its replication.

## FISH assays confirmed that MVM localized with cellular sites of DNA damage

We next sought to confirm the association of MVM replication with sites of cellular DNA damage using super-resolution (STORM) microscopy. For these assays, we designed PCR-based FISH probes complementary to the MVM genome, to a VAD regions at 19qA, and to a control, VAD-negative, site at 6 pA (*Figure 5A*). 3D-FISH combined with confocal imaging of multiple nuclei was performed at 16 hpi. Representative examples are shown in *Figure 5B and D*, which demonstrated close localization between the MVM genome and 19qA-VAD probes (represented by red and green probes respectively), in contrast to the lack of direct localization of MVM with the control probe at chromosome 6 pA (*Figure 5C and D*; represented by red and cyan foci respectively). The 3D distances between the VAD probe and MVM genome in multiple nuclei were calculated using confocal imaging. As shown in *Figure 5E*, the median distance between the MVM genome and the 19qA-VAD and 15qE-VAD probes were approximately 0.7, and 0.6 μm, respectively. This is similar to the median radius of Type II APAR bodies, suggesting that on average VAD sites coincide with APAR bodies. In contrast, non-VAD control sites on chromosome 12 and 17 (12qA3 and 17qA2) exhibited a much greater range of co-location and were separated from its nearest MVM genome by median distances of 1.1 and 1.1 μm, respectively. Taken together, our representative super resolution imaging and quantitative 3D-FISH analyses support V3C results demonstrating that MVM localized with VADs.

## MVM associated with artificially-engineered sites of DNA damage

If MVM preferentially associates with cellular sites of DNA damage, one might expect that MVM could be targeted to artificially–engineered sites of cellular DNA damage. We tested this in two ways. First, we used laser micro-irradiation of MVM-infected A9 cells at 18 hpi to induce focused cellular DNA damage, which is evident as a γ-H2AX 'stripe' in the nucleus (*Figure 6A*). Anti-NS1 staining of these cells suggested that MVM distinctly co-localized with irradiation-induced damaged cellular DNA, and in doing so, viral replication centers adapted to the shape of the damaged DNA stripe (*Figure 6A*, top two panels), rather than the distinct foci characteristic of APAR bodies (representative example shown in *Figure 6A*, bottom panel). Localization of the cellular transcription factor

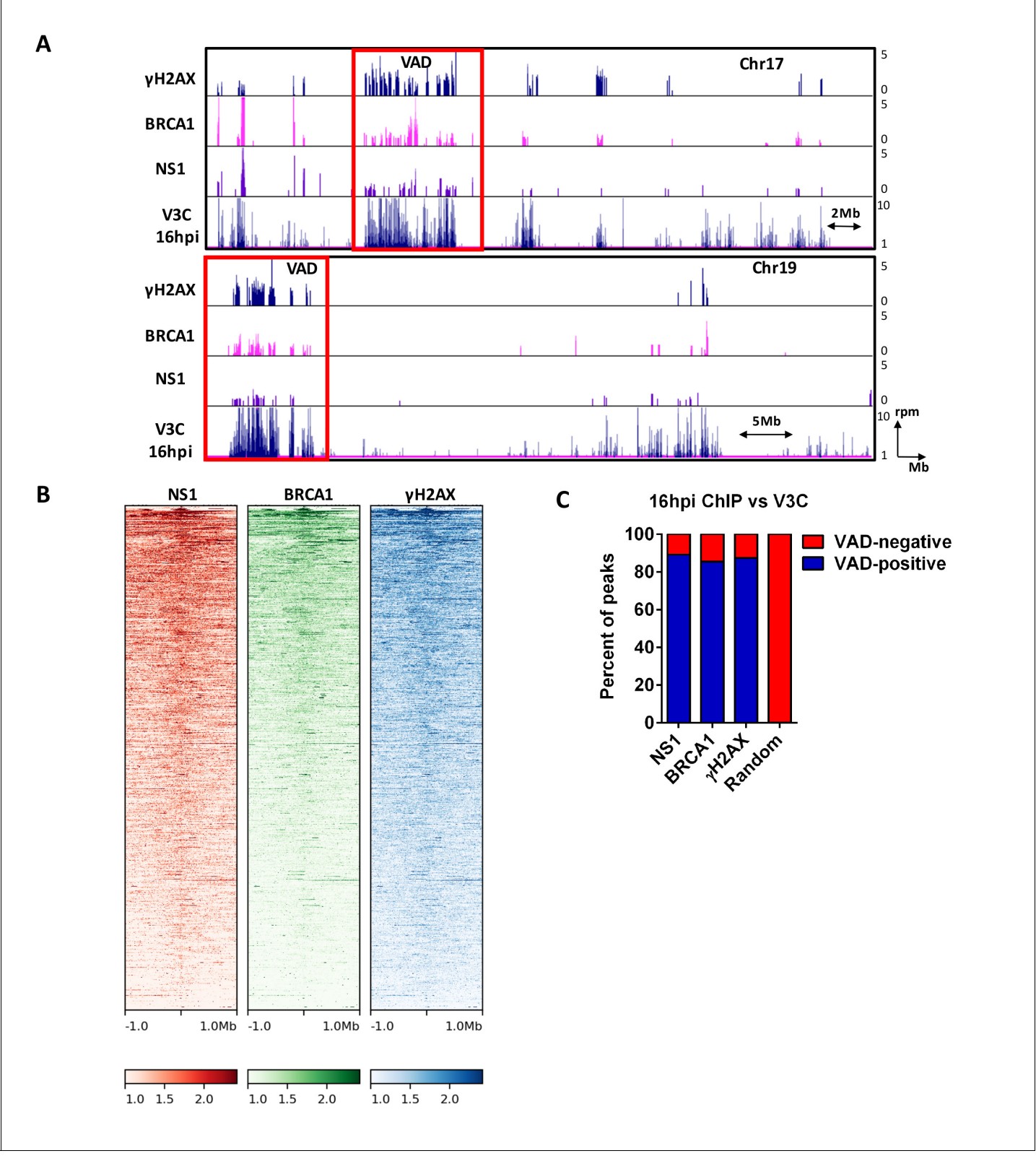

**Figure 4.** MVM NS1 colocalizes to sites of cellular DNA damage along with MVM genome. (**A**) Representative quantile normalized ChIP-seq plots of MVM-NS1 (purple) and BRCA1 (pink) binding to the cellular genome on Chromosome 17 (top) and Chromosome 19 (bottom), with γ-H2AX and V3C at 16 hpi. Red rectangle denotes VAD sites, and y-axis values for ChIP-seq peaks have been restricted from 0 to 5 reads per million. (**B**) The enrichment of NS1 (left), BRCA1 (middle) and γ-H2AX (right) around MVM-associated regions were calculated and plotted as heatmaps using DeepTools on the

*Figure 4 continued on next page*

*Figure 4 continued*

Galaxy server (*Ramírez et al., 2016*). (C) The fraction of NS1 and DDR-positive genomic regions that colocalized with V3C at 16 hpi were calculated using BEDTools, and presented as VAD-positive sites. A library of randomly generated ChIP-seq peaks on the mouse genome with the same fragment size as the called peaks was used as control and also intersected with the MVM-VADs.

DOI: https://doi.org/10.7554/eLife.37750.010

The following figure supplements are available for figure 4:

**Figure supplement 1.** Called peaks from DDR ChIP-seq in HU-treated primary mouse splenocytes were compared with DDR ChIP-seq in MVM infected murine A9 fibroblasts at 16 hpi.

DOI: https://doi.org/10.7554/eLife.37750.011

**Figure supplement 2.** Comparisons of MVM-associated VAD sites with topological structure of the mouse nucleome.

DOI: https://doi.org/10.7554/eLife.37750.012

NR5A2, which is not found in MVM replication centers (*Figure 1B*), was not affected by micro-irradiation (*Figure 6A*, third panel).

These experiments demonstrated that at least NS1 localized to induced sites of DNA damage. The V3C-seq/NS1 ChIP-seq experiments described above suggested that NS1 was a useful surrogate for virus replication in our assays; however, we chose to further assess the direct interaction of the MVM genome with sites of artificially induced cellular DNA damage using directed cleavage of the genome by CRISPR/Cas9 (*Sanjana et al., 2014*). Guide RNAs were designed that targeted a gene desert in chromosome 9 (*Figure 6B*, cytogenetic location at 9qE1). These guides were transfected into A9 fibroblasts stably expressing CRISPR/Cas9. When these cells were then infected with MVM and assayed by focused 3C-qPCR, we detected substantially more amplicons between MVM and the DNA break site in cells transfected with 9qE1 desert-specific guide RNAs, compared to scrambled control guides (*Figure 6C*). Interaction was further confirmed using complementary TaqMan probes that recognize the chromosome 9 CRISPR cleavage site (*Figure 6D*). As expected, MVM interaction with a previously identified VAD on Chromosome 19 (19qA) was not significantly affected in this assay (*Figure 6E*). As an independent verification of the localization of MVM to this site, we also detected NS1 binding to the induced damage site using ChIP, suggesting that NS1 bound to the MVM genome at the break site is secondarily crosslinked and detected by ChIP-qPCR (*Figure 6F*). NS1 binding to the chromosome 19 VAD at 19qA was unaffected in these experiments (*Figure 6G*), consistent with our V3C findings in *Figure 6C and D*.

## Discussion

When DNA viruses enter the nucleus they must locate to sites suitable to sustain replication. Small DNA viruses, such as parvoviruses, require multiple cellular factors for the expression and replication of their genomes. It may be that DNA viruses set up replication centers essentially randomly, and factors necessary for replication are recruited to these sites. An alternative model, suggested by the present work, is that incoming DNA viruses can initially locate to cellular sites that maintain factors necessary for virus replication. Sites of cellular DNA damage present such an opportunity (*Hashiguchi et al., 2007*; *Polo and Jackson, 2011*). Using a high-throughput conformational capture assay developed here for use in trans, we show that at early times post infection MVM interacted directly with sites of cellular DNA damage that in mock infected cells also exhibited DNA damage upon entry into S-phase. As infection progressed, interaction of the MVM genome, as well as the presence of the viral replication protein NS1, increased at these sites, suggesting they were sites of ongoing viral replication. ChIP-seq analysis for γ-H2AX and BRCA1 demonstrated that DNA damage increased during infection, and MVM subsequently associated also with newly induced sites. This model is supported by our findings that MVM NS1 and the MVM genome could be re-localized to artificially induced sites of cellular DNA damage. These observations support a model consistent with the notion that MVM initially establishes replication at cellular DNA damage sites that provide replication and expression machinery, as well as other DDR factors, and as infection progresses induces additional sites of cellular DNA damage, using these to amplify infection. It should be noted, however, that even at the earliest time points examined, multiple virus genomes have accumulated at sites of replication. Thus, specific localization of input genomes remains circumstantial. Consistent with our overall model, we have previously demonstrated that an ATM inhibitor applied during

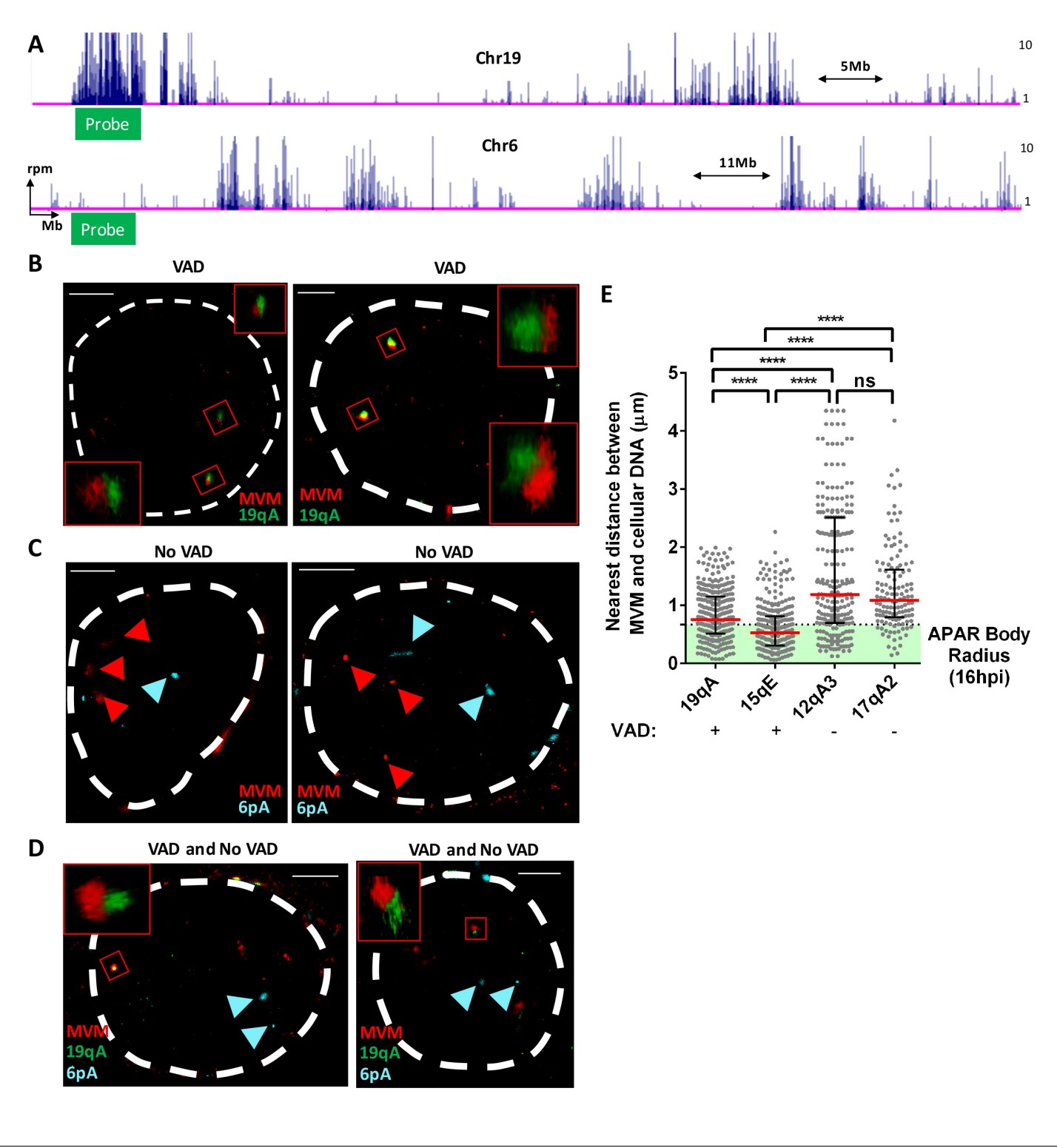

**Figure 5.** FISH assays confirmed that MVM localized with cellular sites of DNA damage. (**A**) Schematic of Chromosomes 19 and 6 showing where MVM associates with the mouse genome in A9 cells at 16 hr post infection, depicting the sites where FISH probes were designed. Representative GSD-STORM images show the spatial localization of the MVM genome (red) with (**B**) chromosome 19 VAD (19qA, green), (**C**) chromosome 6 control site (6 pA, cyan), and (**D**) both probe sets. (**B**) The MVM probe is labelled in red, while the cellular VAD site at 19qA is labelled in green. The insets show magnified views of the MVM-VAD probe sets demarcated by red rectangles in the original image. (**C**) The MVM probe is labelled in red and cellular site associated with non-VAD at 6 pA (shown in 5A, bottom) is labelled in cyan. Red arrowheads indicate the location of the MVM probes, and the cyan
*Figure 5 continued on next page*

*Figure 5 continued*

arrowhead indicates the location of the 6 pA region of the cellular genome. (**D**) Relative locations of the MVM genome with cellular VAD-associated site at 19qA (green) and non-VAD site at 6 pA (cyan) in the same cell. The inset shows a magnified view of the MVM-VAD probe set, and cyan arrows indicate the non-VAD-associated probes. Nuclear borders are labelled in white dotted line. Scale is presented as a white line and measures 2 µm. (**E**) The absolute distance between MVM and cellular genome at sites identified as VADs (19qA and 15qE) versus VAD-negative sites (12qA3 and 17qA2) were calculated using 3D-FISH. Results are depicted as grey dots for each individual APAR body in multiple fields from least three independent infections of parasynchronized A9 cells at 16 hpi, with the median distance represented by a red line. Black error bars represent the interquartile range of the dataset. The median radius of an APAR body at 16 hpi is shown as a dashed horizontal line, and teal shading indicates the domain which would be occupied by the APAR body. Significant differences are denoted as ****p<0.0005 (one-way ANOVA, multiple comparisons). ns designates non-significant statistical differences between datasets.

DOI: https://doi.org/10.7554/eLife.37750.013

infection specifically reduced virus replication (*Adeyemi et al., 2010*). Also, as might be expected, we find that HU pretreatment of permissive rat F111 cells, which have lower levels of endogenous DNA damage as indicated by lower levels of γ-H2AX, resulted in increased MVM replication by 20 hpi (*Figure 3—figure supplement 2B*).

Because VADs correlated strongly with γ-H2AX ChIP-seq sites identified on mock-infected cells as they progressed through S-phase, and on non-infected, HU-treated A9 cells, the predominantly identified γ-H2AX signal associated with MVM during infection very likely resides on cellular DNA. However it remains possible that γ-H2AX and/or other DDR signaling proteins are directly associated with the viral genome during infection. In this regard, while there is a clear potential role in MVM replication for DNA polymerase-δ and gene expression factors potentially present at DNA damage sites, the possible roles of other DDR proteins in parvovirus replication warrants additional study.

Recently Shah and O'Shea have elegantly demonstrated a bipartite cellular DDR to adenovirus (Ad) infection that initially targets the virus while sparing the cell. The Mre11-Rad50-Nbs1 (MRN) complex first inhibits adenovirus replication without inducing a global response that could interfere with cellular proliferation or viability. As Ad overcomes this block, utilizing the Ad E4Orf6/E1B 55 kDa complex, and replicates to high levels, a global DDR is subsequently induced, and cellular DNA breaks were found to sequester DDR proteins from adenovirus preventing its replication (*Shah and O'Shea, 2015*). While MVM localizes to sites of cellular DNA damage as replication ensues, the model that we propose for parvovirus infection is different. It is consistent with the parvovirus life cycle, which depends upon the induction of a cell cycle arrest, is susceptible to ATM inhibitors, and which continually induces cellular DNA damage during its infection. Additionally, in contrast to adenovirus, MVM is much more dependent on host cell factors for its replication and expression (*Cotmore and Tattersall, 2014*), and it does not have an extensive genetic capacity to encode functions that inactivate the cellular DDR as does adenovirus (*Ou et al., 2012*; *Querido et al., 2001*; *Stracker et al., 2002*).

Chromosome conformation assays have been traditionally used to analyze the 3D folding principles of the cellular genome, including the regulation of promoter-enhancer loops and structural folding loops (*Dixon et al., 2016*). However, these assays can also serve as valuable tools to study the interaction between the host genome and invading virus. In the context of virus-infected cells, Chromatin Interaction Analysis by Paired-End Tag Sequencing (ChIA-PET) assays have enabled the comprehensive mapping of EBV interaction with the cellular genome (*Jiang et al., 2017*). These studies, which combine ChIP-seq with chromosome conformation capture techniques, utilize the proximal interaction of distally located DNA regions bound by shared protein elements. They have demonstrated that Epstein Barr Virus enhancers can regulate the expression of the cellular Myc oncogene in lymphoblastoid cells via long-range promoter-enhancer looping, thereby contributing the EBV-mediated cellular transformation (*Jiang et al., 2017*). However, mapping of the virus-host interactome by ChIA-PET experiments can be limited by the necessity of having *a priori* knowledge of the proteins mediating this interaction. The EBV interactome has also been characterized using in-situ Hi-C assays, which generate chromosome conformation maps that provide a snapshot of how every restriction enzyme site associates with every other site throughout the genome. These studies showed that the latent EBV episome associates with gene poor regions, but relocalizes to gene-rich regions of the genome upon reactivation (*Moquin et al., 2017*). While highly informative, these studies did not have the resolving power of the assays described here for MVM.

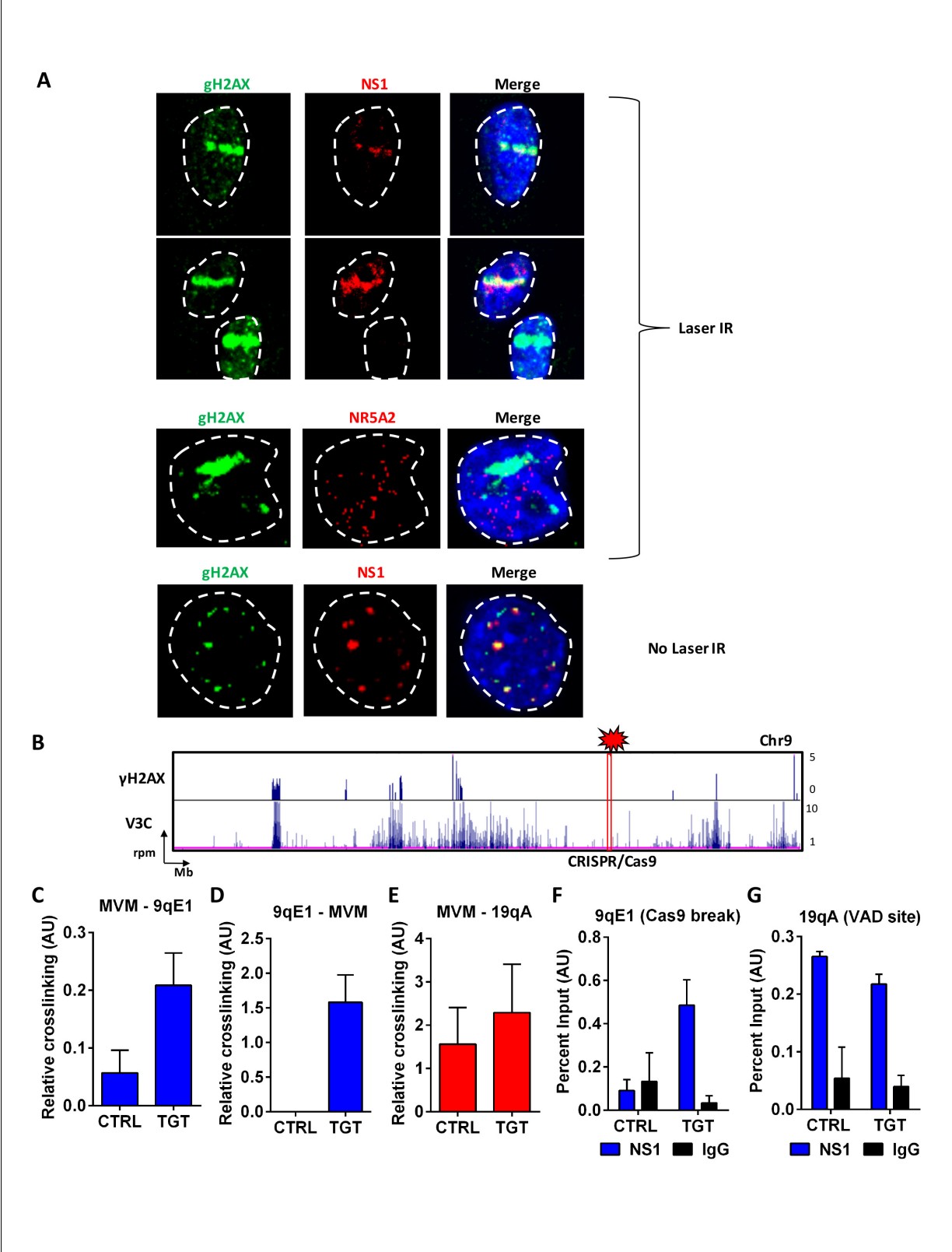

**Figure 6.** MVM associates with artificially-engineered sites of DNA damage. (**A**) Murine A9 fibroblasts were infected with MVM at an MOI of 10 for 18 hr. Cells were sensitized with Hoechst 33342 solution prior to irradiation in selected regions of interest with a 405 nanometer laser (top 3 panels), and were compared with un-irradiated cells (bottom panel). Cells were fixed and analyzed for the localization of DNA damaged sites (detected by green γ-H2AX staining), with that of MVM NS1 protein (*red*). The irrelevant transcription factor NR5A2 (*red*) was used as a negative control as this marker that

*Figure 6 continued on next page*

*Figure 6 continued*

does not colocalize with APAR bodies (see *Figure 1*). The nuclear periphery is demarcated with dotted white lines and DAPI staining. (**B**) Schematic of γ-H2AX binding to the cellular genome on chromosome 9 (top panel) compared with MVM-interaction sites (bottom panel) in parasynchronized murine A9 cells infected with MVM (MOI 5) for 16 hr. The site where guide RNAs were designed for DDR induction is shown as a red rectangle and labelled 'CRISPR/Cas9'. Focused V3C-qPCR assays were performed in A9 cells constitutively expressing LentiCRISPRv2 which were transiently transfected with guide RNAs (nontargeting, labelled as 'CTRL' or specific to the 9qE1 site, labelled as 'TGT'), and infected with MVM at an MOI for 5 for 16 hr. The spatial association of MVM with cellular sites were determined with Taqman probes on (**C**), the MVM genome, and reciprocal interaction with the Taqman probe on (**D**), the 9qE1 break site. (**E**) MVM association with Chr19 VAD at 19qA was tested by focused Taqman qPCR. NS1 occupancy at (**F**), the 9qE1 break site and (**G**), the 19qA VAD was assayed by ChIP-qPCR in cells transfected with non-targeting (CTRL) and specific (TGT) guide RNAs. Background levels were determined using IgG pulldowns. qPCR data are presented as mean ± SEM of three independent experiments.

DOI: https://doi.org/10.7554/eLife.37750.014

For MVM, a single inverse-PCR viewpoint was sufficient to map its interactome. Moreover, using inverse PCR using primers complementary to the viral genome to generate the sequencing library ensured that the V3C-seq assay detects only MVM-host hybrid junctions. This enabled higher resolving power and a deeper interrogation of genomic associations, allowing the detection of both frequent and infrequent MVM interaction sites. The drawback of V3C-seq, however, is that it does not assay changes in nuclear architecture in response to viral infection, which invariably is altered during parvovirus replication, particularly in late stages.

Looping interactions have previously been observed at distinct loci in genomic regions proximal to some integrated viral genomes. For example, the integrated Murine Leukemia Virus (MuLV) genome has beRen, BRen, Ben shown to influence the folding properties of its proximally located Myc promoter, which in turn contributes to cellular oncogenesis (*Zhang et al., 2012*). In other studies, it has been shown in cell line models that the integrated HIV genome associates distally with an uncharacterized chromosomal region to promote reactivation of latent HIV (*Dieudonné et al., 2009*). Chromosome conformation capture analyses focused on elucidating the topological conformation of the viral genome have also revealed the conformational structure adopted by the gamma-herpesvirus Kaposi's Sarcoma Associated Herpesvirus (KSHV), which forms distinct promoter-enhancer loops mediated by the cellular architectural proteins CTCF and Cohesin (*Kang et al., 2011*). In this study we have utilized chromosome conformation capture technology for the first time to map the *trans* interaction of a lytic virus with the cellular genome in a non-biased way. However, in addition to assaying the inter-chromosomal interaction cellular sites, long-range interactions of integrated viruses such as MuLV and HIV with distal elements, or TADs, may also be assayed using V3C-seq.

Inspection of VADs show that regions of MVM interaction vary greatly in size. The size of the peaks correlates with the number of interactions, and so indicate that some regions are more densely populated by virus than others. As can be seen in the whole-genome analysis, there is quite a variation in VADs across different chromosomes (*Figure 2—figure supplement 2*). While most chromosomes have multiple VADS, most have only a few larger VADs of 5–15 Mb. Whether the size of the VADs correlates to the success of infection at that site is not known. There are a number of chromosomes [chromosomes 1, 13, 14, 18, X and Y (*Figure 2—figure supplement 2* and *Figure 3—figure supplement 1*)] that have few sites of damage, as assessed by γ-H2AX ChIP-seq, as well as few VADs, reinforcing their correlation.

VADs identified by V3C-seq showed striking overlap with sites of damage in infected cells, at sites of damage incurred in uninfected cells as they progressed into S-phase, but surprisingly, also at sites in uninfected cells treated with HU. This overlap suggested that there may be a predilection in the cellular chromosome for sites sensitive to the induction of damage for which MVM has an affinity. In this vein, we find that most VADs (and by implication sites of endogenous and infection-induced damage and damage induced by HU) also overlap strikingly with Topologically Associated Domains (TADs), determined for murine CH12 B-cells, by Hi-C assays (*Figure 4—figure supplement 2A–D*, [*Rao et al., 2014*]). Such studies, in combination with high-resolution imaging, have shown that the mammalian genome is folded into megabase-sized cis-interacting regions (TADs). TADs have been divided into two main compartments: A (primarily euchromatin-like), and B (primarily heterochromatin-like) (*Rao et al., 2014*). The A compartment is characterized by being gene dense, containing highly expressed genes, and active chromatin marks. The A1 subcompartment was seen to finish

replicating at the beginning of S-phase, while the A2 subcompartment was described as continuing to replicate into the middle of S-phase. Overlaying published chromatin marks from murine 3T3 cells induced with the DNA damaging agent aphidicolin (*Kraushaar et al., 2013*) suggests that MVM associations may be with the A compartment (*Figure 4—figure supplement 2E*). This compartment of cellular DNA would provide the necessary machinery for the expression and replication of MVM within S-phase, and would be consistent with MVM initially associating with sites that emerge as ERFs during cellular replication, which would be predicted to replicate in compartment A.

The VADs identified in our study occupy multiple adjacent TADs. This suggests that several adjacent TADs may contain the necessary environment to support MVM expression and replication. The mechanisms that maintain TAD borders are elusive, but structural proteins such as CTCF and cohesin have been implicated in these processes (*Dixon et al., 2016*; *Rao et al., 2014*, *2017*). Indeed, Canela et al., have shown that CTCF and RAD21 form the anchors to cis-interacting cellular loops that are susceptible to topoisomerase TOP2B mediated DNA breaks (*Canela et al., 2017*). The MVM NS1 protein, which also interacts at TADs, is a known DNA binding protein with double-stranded nickase activity; however, the mechanisms by which the viral genome associates with the cellular sites remains to be determined.

Our results suggest that host chromatin states may play a significant role in permissiveness to DNA damage, and thereby influence MVM localization for replication. Consistent with this model, the HPV E2 protein has been found associated with actively transcribing genes and active chromatin, presumably to facilitate expression of HPV through hijacking the host transcriptional machinery at these sites (*Jang et al., 2009*). Indeed, profiling of sites on the cellular genomic that undergo replication stress have shown that these regions are encased in a protective chromatin environment to facilitate efficient repair (*Kim et al., 2018*). Such regions, either pre-existing prior to infection, or induced by virus, may provide the supportive environment necessary for successful infection. The generally adaptable, high-throughput V3C-seq assay allows genome-wide identification of the direct associations of viral genomes with distinct regions of the cellular chromosome. It should be useful for characterizing the interaction of many DNA viruses that associate with the cellular genome, and thus provide a useful tool to characterize the molecular events leading to the initiation of their infections.

# Materials and methods

**Key resources table**

| Reagent type (species) or resource | Designation | Source or reference | Identifiers | Additional information |
|---|---|---|---|---|
| Cell line (*Mus musculus*, Male) | A9 | ATCC, (*Tattersall and Bratton, 1983*) PMID: 6602222 | RRID:CVCL_3984 | Verified as mycoplasma-negative by PCR |
| Cell line (*Mus musculus*) | NIH-3T3 | ATCC | RRID:CVCL_0594 | Verified as mycoplasma-negative by PCR |
| Cell line (*Homo sapiens*) | NB-324K | (*Tattersall and Bratton, 1983*) PMID: 6602222 | RRID:CVCL_U409 | Verified as mycoplasma-negative by PCR |
| Cell line (*Mus musculus*) | EL4 | ATCC, (*Tattersall and Bratton, 1983*) PMID: 6602222 | RRID:CVCL_0255 | Verified as mycoplasma-negative by PCR |
| Cell line (*Rattus norvegicus*) | F111 | Fischer Rat Fibroblasts; (*Freeman et al., 1975*) | RRID:CVCL_6C52 | Verified as mycoplasma-negative by PCR |
| Antibody | NS1 | Salome and Pintel, unpublished | 2C9b | anti-mouse; Usage per sample: ChIP: 6 µg Immunofluorescence: 2 µg IB: 2 µg |
| Antibody | γ−H2AX | EMD Millipore | EMD Millipore:05–636 | anti-mouse; Usage per sample: ChIP: 5 µg |
| Antibody | γ−H2AX | Abcam:ab11174 | RRID:AB_297813 | anti-rabbit; Usage per sample: ChIP: 5 µg Immunofluorescence: 2 µg IB: 2 µg |

*Continued on next page*

*Continued*

| Reagent type (species) or resource | Designation | Source or reference | Identifiers | Additional information |
|---|---|---|---|---|
| Antibody | BRCA1 | Thermo Fisher Scientific:17F8 | RRID:AB_557804 | anti-mouse; Usage per sample: ChIP: 5 μg |
| Antibody | FANCD2 | Bethyl Laboratories: a302-174A | RRID:AB_1659803 | anti-rabbit; Usage per sample: Immunofluorescence: 2 μg |
| Antibody | NR5A2 | Abcam:ab189876 | RRID: AB_2732890 | anti-rabbit; Usage per sample: Immunofluorescence: 2 μg |
| Antibody | IgG | Cell Signaling | Cell Signaling:5415S | mouse; Usage per sample: ChIP: 5 μg |
| Antibody | AF 488 | Life Technologies: A11034 | RRID:AB_2576217 | anti-rabbit secondary; Usage per sample: Immunofluorescence: 1 μg |
| Antibody | AF 568 | Life Technologies: A11031 | RRID:AB_144696 | anti-mouse secondary; Usage per sample: Immunofluorescence: 1 μg |
| Antibody | AF 555 | Life Technologies: A27039 | RRID:AB_2536100 | anti-rabbit secondary; Usage per sample: Immunofluorescence: 1 μg |
| Antibody | AF 647 | Life Technologies: A32728 | RRID:AB_2633277 | anti-mouse secondary; Usage per sample: Immunofluorescence: 1 μg |
| Recombinant DNA reagent | Lenti-CRISPRv2 | Addgene; (*Sanjana et al., 2014*) PMID: 25075903 | Addgene plasmid 52961 | |
| Recombinant DNA reagent | pgRNA-humanized | Addgene; (*Qi et al., 2013*) PMID: 23452860 | Addgene plasmid 44248 | |
| Peptide, recombinant protein | HindIII | New England Biolabs | NEB:R0104 | |
| Peptide, recombinant protein | NlaIII | New England Biolabs | NEB:R0125 | |
| Peptide, recombinant protein | T4 DNA Ligase | New England Biolabs | NEB:M0202 | |
| Commercial assay or kit | FISH Tag DNA Multicolor Kit | Thermo Fisher Scientific | Thermo Fisher Scientific: F32951 | |
| Commercial assay or kit | QIAquick PCR purification kit | Qiagen | Qiagen:28106 | |
| Commercial assay or kit | NEBNext Ultra II Library Prep Kit for Illumina | New England Biolabs | NEB:E7645 | |
| Commercial assay or kit | Trevigen Comet Assay Kit | Trevigen | Trevigen: 4250–050 K | |
| Commercial assay or kit | StemCell | EasySep Human CD4+ T Cell Enrichment Kit | StemCell:19052 | |
| Chemical compound, drug | Hydroxyurea | Sigma Aldrich | Sigma Aldrich:H8627 | |
| Chemical compound, drug | Doxorubicin | Sigma Aldrich | Sigma Aldrich:D1515 | |
| Chemical compound, drug | Bovine Serum Albumin | Sigma Aldrich | Sigma Aldrich:A2153 | |
| Chemical compound, drug | ProLong Diamond Antifade Mountant with DAPI | Thermo Fisher Scientific | Thermo Fisher Scientific: P36966 | |
| Chemical compound, drug | Hoechst 33342 | Thermo Fisher Scientific | Thermo Fisher Scientific: 62249 | |
| Software, algorithm | Bowtie2 | (*Langmead and Salzberg, 2012*) PMID: 22388286 | | http://bowtie-bio.sourceforge.net /bowtie2/index.shtml |

*Continued on next page*

*Continued*

| Reagent type (species) or resource | Designation | Source or reference | Identifiers | Additional information |
|---|---|---|---|---|
| Software, algorithm | Samtools | (*Li et al., 2009*) PMID: 19505943 | RRID:SCR_006646 | http://samtools.sourceforge.net/ |
| Software, algorithm | Bedtools | (*Quinlan and Hall, 2010*) PMID: 20110278 | RRID:SCR_006646 | http://bedtools.readthedocs.io/en/latest/ |
| Software, algorithm | Deeptools | (*Ramírez et al., 2016*) PMID: 27079975 | RRID:SCR_016366 | https://deeptools.readthedocs.io/en/develop/ |
| Software, algorithm | UCSC Genome Browser | (*Kent et al., 2002*) PMID: 12045153 | RRID:SCR_005780 | https://genome.ucsc.edu/ |
| Software, algorithm | PreprocessCore | (*Bolstad, 2013*) | | https://www.bioconductor.org/packages/release/bioc/html/preprocessCore.html |
| Software, algorithm | Biostrings | (*Pagès et al., 2017*) | | https://bioconductor.org/packages/release/bioc/html/Biostrings.html |
| Software, algorithm | EPIC | (*Xu et al., 2014*) PMID: 24743992 | | https://github.com/biocore-ntnu/epic |
| Software, algorithm | Galaxy | (*Afgan et al., 2016*) PMID: 27137889 | RRID:SCR_006281 | https://usegalaxy.org/ |
| Software, algorithm | ImageJ | (*Schneider et al., 2012*) PMID: 22930834 | RRID:SCR_003070 | https://imagej.net/Welcome |
| Software, algorithm | Huygens Professional | Huygens professional version 17.10 (Scientific volume imaging, The Netherlands) | | https://svi.nl/Huygens-Professional |

## Contact for reagent and resource sharing

Further information and requests for resources and reagents should be directed to and will be fulfilled by David Pintel (pinteld@missouri.edu).

## Experimental models and subject details

Cell lines were cultured in 5 percent FBS-containing DMEM media (5 percent $CO_2$ and 37 degrees Celsius). Murine EL4 cells were cultured in RPMI media with 5 percent FBS. Cell lines are routinely authenticated for mycoplasma contamination, and background levels of DNA damage detected by γ-H2AX. Further information on cell line authentication and parvovirus replication are available at ATCC and published studies (*Tattersall and Bratton, 1983*) respectively.

## Cell lines, viruses and viral infection

Male Murine A9, NIH-3T3, EL4 and human NB324K cells were propagated and wild-type MVMp and MVMi were produced as previously described (*Adeyemi et al., 2010*). Infection was carried out at a Multiplicity Of Infection (MOI) of 5 unless otherwise stated, leading to infection rates of 70–80% as detected by NS1 staining.

## LentiCRISPRv2 A9 cells

LentiCRISPRv2 plasmid was obtained from Addgene (plasmid# 52961, [*Sanjana et al., 2014*]) and pseudotyped viruses were generated in $1 \times 10^6$ 293 T cells transfected with 1 μg of LentiCRISPRv2, 1 μg of HIV Gag/Pol and 1 μg VSV-G proteins using Lipo293D (SignaGen). Supernatant containing the lentivirus was collected at 48 hr post-transfection. Independent preparations of LentiCRISPRv2 lentivirus were used to transduce A9 cells with 750 μl of lentiviral supernatant for 48 hr before selecting cells in 1 μg/ml of puromycin for 10 days. The resulting polyclonal puromycin-selected LentiCRISPRv2 A9 cells were validated for Cas9 expression by western blot, and were utilized for induced DNA break assays (described below).

## Methods details

### Cell synchronization and drug treatments

Murine A9 and human NB324K cells were parasynchronized in G0 phase of cell cycle by growing them in 5% FBS containing DMEM without isoleucine for 36–42 hr (as described previously, (*Adeyemi et al., 2010*). NIH-3T3 cells were parasynchronized in G0 phase of cell cycle by growing them in 0.5% FBS containing DMEM for 36 hr. All cells were released into complete media containing 5% FBS in DMEM, and infected with MVMp at the time of release. Entry into S phase of cell cycle occurs approximately 8–10 hr after release into complete media. 16 hr post infection thus represents approximately 8–10 hr of transit into S-phase. Virally infected cells were harvested at the indicated timepoints and processed for experiments.

### Chromosome conformation capture (3C) assay

Chromosome Conformation Capture assays were performed using $10^7$ cultured A9, EL4, NIH-3T3 and NB-324K cells. Briefly, samples were cross-linked in 2 percent formaldehyde for 10 min, before quenching them in 0.125 M glycine. Cells were lysed in NP40 lysis buffer (0.1% NP40, NaCl, Tris-HCl) and the resulting nuclei were resuspended in restriction enzyme buffer (NEB Buffer 2.1). The nuclei were permeabilized in 0.3% SDS for an hour, followed by sequestration of SDS in 2% Triton X-100. The samples were digested in 400U of Hind III restriction enzyme overnight. Digestion was continued with a further 300U of Hind III on the next day, before inactivating the enzyme with 1% SDS at 65°C. SDS was sequestered with 1% Triton X-100, and 3C chromatin was resuspended in 1.15X T4 DNA Ligase Reaction Buffer. 50U of T4 DNA Ligase was added to the samples. Intramolecular ligation was carried out at room temperature for 4 hr, before reversing the crosslinks and digesting protein at 65 degrees C overnight with Proteinase K. 3C DNA was purified by phenol: chloroform:isoamyl alcohol extraction, isopropanol precipitation and finally using a PCR purification kit. The 3C-DNA was eluted in 200 microliters of Buffer EB (Qiagen). Cross-linking efficiencies were measured using Taqman-qPCR assays with primers and probes shown in *Supplementary file 1*. Relative crosslinking between two distally located HindIII fragments was determined by the ratio of the novel ligation junction to that of nearest neighbor interaction on the *Ercc3* locus, as described previously (*Hagège et al., 2007*).

### Viral chromosome conformation capture sequencing (V3C-seq) assay

V3C-seq assays were performed with Hind III as the primary restriction enzyme to digest cross-linked MVM infected A9 fibroblast chromatin. The Hind III-digested DNA was intramolecular-ligated using the 3C procedure, before resuspending in Buffer EB (100 μl, Qiagen). 3C-DNA was secondary-digested with Nla III (100U, overnight at 37°C), before being heat inactivated and circularized with 100U of T4 DNA Ligase at room temperature overnight in 6 ml of ligation reaction. The V3C samples were precipitated by phenol:chloroform extraction, precipitated in isopropanol, resuspended in Qiagen Buffer EB (100 μl), and. Inverse PCR was performed on the circularized DNA using primers within the Hind III - Nla III fragments on the MVM genome using inverse PCR primers described in *Supplementary file 1*. Inverse PCR products were diluted 1:100 in TE buffer and used as templates for nested inverse PCRs (described in *Supplementary file 1*), yielding V3C-seq DNA libraries. Sequencing libraries were prepared using the NEB Ultra Kit, and twelve samples were pooled per run for 75 base-pair single end sequencing using an Illumina Next Seq 500 sequencer.

### V3C-seq analysis

V3C-seq samples were trimmed and aligned to the mouse reference genome (mm10 build) using Bowtie2 (*Langmead and Salzberg, 2012*). The Biostrings package in RStudio was used to generate a genome-wide map of HindIII restriction fragments for the assignment of reads (*Pagès et al., 2017*). To compare between different timepoints, reads for each fragment were averaged and quantile normalized using preprocessCore package on RStudio (*Bolstad, 2013*). For visualization of the V3C-seq data, a running mean was calculated using a window size of five contiguous HindIII fragments (*Medvedovic et al., 2013*). Bioinformatic codes provided in *Table 1*.

**Table 1.** Bioinformatic codes used.

| Program | Function | Code used |
|---------|----------|-----------|
| V3C-seq analysis | | |
| Bowtie2 | Alignment | bowtie2 –trim5 50 –very-sensitive -x/storage/htc/biocompute/ircf/dbase/genomes/M_musculus/bowtie2/index/mm10 -S 24hpi_1.sam 24hpi_1.fastq |
| Samtools | Sam to Bam | samtools view -b -S -o aligned_24hpi_1.bam 24hpi_1.sam |
| | Sort | samtools sort -o aligned_sorted_24hpi_1.bam aligned_24hpi_1.bam |
| BEDtools | Compute histogram | genomeCoverageBed -ibam aligned_sorted_24hpi_1.bam -bg -trackline -split -g. ..>24hpi_1.bedgraph |
| ChIP-seq analysis | | |
| Bowtie2 | Alignment | bowtie2 -x/storage/htc/biocompute/ircf/dbase/genomes/M_musculus/bowtie2/index/mm10 -U 16hpi_gh2ax_1.fastq -S 16hpi_gh2ax_1.sam |
| Samtools | Sam to Bam | samtools view -b -S -o aligned_16hpi_gh2ax_1.bam 16hpi_gh2ax_1.sam |
| | Sort | samtools sort -o aligned_sorted_16hpi_gh2ax_1.bam aligned_16hpi_gh2ax_1.bam |
| BEDtools | Bam to BED conversion | bedtools bamtobed -i 16hpi_gh2ax_1.bam>16hpi_gh2ax_1.bed |
| EPIC | Peak Calling | epic -t 16hpi_gh2ax_1.bed -c 16hpi_ip.bed -gn mm10 -b BED -o epic_12hpi_gh2ax_1 |
| BEDtools | Intersection | bedtools intersect –a 16hpi_gh2ax_1.bed –b 16hpi_gh2ax_2.bed>16hpi_gh2ax_1_2.bed |
| | Jaccard analysis | bedtools jaccard -a mock_gh2ax.bed -b 16hpi_gh2ax.bed |

DOI: https://doi.org/10.7554/eLife.37750.015

## Chromatin immunoprecipitation (ChIP) assay

The indicated cells were cross-linked with 1% formaldehyde for 10 mins at room temperature and then quenched with 0.125 M glycine. The cells were collected and lysed using a ChIP lysis buffer (1% SDS, 10 mM EDTA, 50 mM Tris-HCl, pH 8, protease inhibitors) for 20 min on ice. The lysates were sonicated using a Diagenode Bioruptor for 75 cycles (30 s on and 30 s off per cycle), before being incubated overnight at 4°C with the indicated antibodies bound to Protein A Dynabeads (Invitrogen), in ChIP dilution buffer (0.01% SDS, 1.1% Triton X-100, 1.2 mM EDTA, 16.7 mM Tris-HCl pH8, 167 mM NaCl). Samples were washed for 3 min each at 4 degrees Celsius with low salt wash (0.01% SDS, 1% Triton X-100, 2 mM EDTA, 20 mM Tris-HCl pH8, 150 mM NaCl), high salt wash (0.01% SDS, 1% Triton X-100, 2 mM EDTA, 20 mM Tris-HCl pH8, 500 mM NaCl), lithium chloride wash (0.25M LiCl, 1% NP40, 1% DOC, 1 mM EDTA, 10 mM Tris-HCl pH8) and twice with TE buffer before being eluted with SDS- elution buffer (1% SDS, 0.1M Sodium bicarbonate). Following elution, the chroma-tin-antibody-DNA complexes, and the input chromatin were subjected to proteinase K treatment at 65°C overnight. The ChIP DNA was purified using a PCR purification kit (Qiagen), and eluted in 100 ul of Buffer EB (Qiagen). ChIP assays were analyzed by quantitative PCR (qPCR) with iTaq universal SYBR green mastermix (Bio-Rad), using primer sets described in *Supplementary file 1*, or sequenced as described below. Percent input was calculated as described previously (*Fuller et al., 2017*).

Sequencing libraries were generated from ChIP DNA using the NEBNext Ultra II Library Prep Kit for Illumina, and the sonication quality was determined using Agilent Bioanalyser. For ChIP-seq, twelve samples were pooled and sequenced on an Illumina Next Seq 500 using 75 base-pair Single End sequencing.

## ChIP-seq analysis

ChIP-seq samples were aligned to the mouse genome (build mm10) using Bowtie2 (*Langmead and Salzberg, 2012*). Peaks were called with EPIC analysis software (using the SICER algorithm (*Zang et al., 2009*) according to default parameters. Called-peaks that were shared between repli-cates were identified using BEDtools software (*Quinlan and Hall, 2010*). Comparison between ChIP-seq and V3C-seq peaks were performed using Deeptools package (*Ramírez et al., 2016*). In order

to compare the magnitudes of ChIP-seq peaks between different timepoints of MVM infection and mock versus Hydroxyurea treatment, rpm values were calculated (using Galaxy, [*Afgan et al., 2016*]) on the bedgraph files generated from EPIC, and were quantile normalized using preprocessCore package on RStudio (*Bolstad, 2013*). Bioinformatic codes provided in *Table 1*.

## Laser Micro-Irradiation assays

Laser micro-irradiation was performed on 1 million A9 cells cultured on glass bottom dishes (MatTek Corp.) infected with MVMp at an MOI of 10 for 18 hr. Cells were sensitized with 2 microliters of Hoechst dye (ThermoFisher Scientific) 5 min prior to irradiation. Samples were irradiated using a Leica TCP SP8 confocal microscope with a 405 nm laser using 25% power at 40 Hz frequency for 2 consecutive frames per field-of-view. Regions of interest (ROIs) were selected within the nucleus without traversing the nuclear membrane. Samples were processed for immunofluorescense imaging without CSK pre-extraction immediately after micro-irradiation.

## CRISPR-Induced DNA break assays

Stable A9 cells expressing LentiCRISPRv2 were co-transfected with guide RNAs targeting chromosome 9 at 9qE1 (labelled as TGT), or scrambled control guide RNAs (labelled as CTRL) and human CD4 expressing vector during parasynchronization. CD4-positive cells were purified using an Easy-Sep CD4+ T Cell Enrichment Kit (StemCell) prior to release into complete DMEM media and MVM infection. Infected cells were harvested and processed for ChIP and 3C assays at the indicated timepoints.

## 3D-FISH assays

The MVMp genome and indicated cellular regions were labelled with the DNA FISH-Tag Multicolor Kit (ThermoFisher). Briefly, 1 µg of DNA was labelled with aminoallyl-modified dNTP by nick-translation using the manufacturer's instructions before being labelled with amine-modified Alexa-Fluor dyes (AlexaFluor 488 and AlexaFluor 555). The dye combinations were resuspended at equimolar amounts in hybridization buffer (50% formamide, 2X SSC, 40% dextran sulfate, 10% Denhardt's solution) prior to hybridizing to the sample.

Parasynchronized MVMp-infected A9 cells were harvested at the indicated timepoints by pre-extracting with CSK Buffer (10 mM PIPES pH 6.8, 100 mM Sodium Chloride, 300 mM Sucrose, 1 mM EGTA, 1 mM Magnesium Chloride) for 3 min followed by CSK Buffer with 0.5% Triton for 3 min. Cells were crosslinked with 4% paraformaldehyde for 10 min at room temperature, before being washed with PBS. The nuclei were dehydrated by sequential treatments with 50% ethanol, 70% ethanol and 100% ethanol for 3 min each. Nuclei were subsequently rehydrated with sequential treatment with 70% ethanol, 50% ethanol and PBS for 3 min each. Cells and nuclei were permeabilized with 0.5% Triton X-100 in PBS for 15 min, before being washed with PBS. The samples were treated with 2 µg of RNAse A (Roche) in PBS for 1 hr at 37°C. Samples were denatured in 50% formamide (Ambion) before being hybridized to the suspended fluorescently labelled probes overnight at 37°C. Samples were washed in 2X SSC with 0.1% Tritox X-100 at 37°C followed by three times for 5 min each, followed by 2 washes in 2X SSC at 37°C for 5 min each. Samples were then mounted on slides and imaged on the indicated microscope.

For GSD/dSTORM super-resolution imaging, coverslips with adherent immunostained cells were mounted on cavity microscope slides with PBS (0.01 M, pH 7.4) imaging buffer containing 100 mM beta-mercaptoethylamine, 10% w/v glucose, 0.5 mg/ml glucose oxidase and 40 µg/ml catalase. GSD super-resolution imaging was performed on a Leica SR GSD 3D microscope (Leica Microsystems, Inc.) using a 560 nm (AlexaFluor 555) or a 632 nm (AlexaFluor 647) excitation lasers and a 405 nm back-pumping (activation) laser. A 160 × 1.43 NA oil-immersion objective lens was used for imaging. Two-color GSD images were acquired sequentially with an Andor iXon Ultra 897 EMCCD camera at exposure times 7–8 ms using a QGSD 561 quad filter cube and emission bandpasses 605/45 nm (AlexaFluor 555) and 695/85 nm (AlexaFluor 647). Approximately 8000 images per channel of a 18 × 18 µm field-of-view were acquired. The coordinates of single molecules were localized in all recorded raw images and high-resolution GSD images were constructed using Leica LAS X software (version 1.9).

For confocal imaging, samples were mounted on slides using Pro-Long Diamond anti-fade media with DAPI (Invitrogen). Confocal z-stacks were acquired using a Leica TCP SP8 confocal microscope with 488 nm (AlexaFluor 488) and 552 nm (AlexaFluor 555) excitation lasers and a 100 × 1.4 NA objective lens.

3D-FISH images were analyzed using ImageJ. Background noise was filtered out using the Kalman Stack Filter plugin to determine the coordinates (x,y,z) of the centers of the foci. The coordinates of the viral and cellular foci were measured using Sync Measure 3D. The 3D-distance was calculated by computing the displacement vector between the two locations as described previously (*Shih and Krangel, 2010*).

### Immunofluorescense assays

Parasynchronized MVMp-infected A9 cells were harvested at the indicated time points processed as described above till permeabilization with 0.5% Triton X-100 in PBS. Samples were blocked with 3% BSA in PBS for 1 hr, incubated with the indicated antibodies for 1 hr, and incubated with the indicated secondary antibodies (tagged with Alexa Fluor fluorophores) for 1 hr. Samples were washed and mounted on slides with ProLong Diamond Antifade Mountant with DAPI (Invitrogen).

### Alkaline comet assay

Alkaline Comet Assays were performed using Trevigen Comet Assay kits. Murine A9 fibroblasts were grown on 10 centimeter dishes and mock infected, induced with Doxorubicin (200 nM) for 9 hr, or infected with MVMp at an MOI of 10 for 20 hr, before detaching them from the flask by scraping. Cells were washed with ice-cold PBS and resuspended at a density of $10^5$ cells/ml in ice cold PBS. Cells were combined with molten LM Agarose at 37°C at a ratio of 1:10 and pipetted onto Comet Slides. Slides were placed at 4°C in the dark for 10 mins. Slides were immersed in 4°C Lysis solution for 30–60 min, before placing in Alkaline Unwinding Solution for 1 hr at 4°C in the dark. 850 ml Alkaline Electrophoresis Solution was added to the slide tray, and 21 Volts were applied for 30 min. Slides were immersed in water twice for 5 min each, followed by immersion in 70% ethanol for 5 min. Samples were dried at 37 degrees Celsius for 15 min, and subsequently stained with 100 ul of SYBR Gold for 30 min in the dark. Slides were briefly rinsed in water and completely dried at 37°C. Slides were imaged on a Leica widefield microscope.

### Immunoblot analysis

Cells grown and infected in 60 mm dishes were harvested at the indicated timepoints, followed by lysis in modified RIPA buffer (20 mM Tris-HCl pH 7.5, 150 mM NaCl, 10% glycerol, 1% NP-40, 1% Sodium Deoxycholate, 0.1% SDS, 1 mM EDTA, 10 mM trisodium pyrophosphate, 20 mM Sodium Fluoride, 2 mM Sodium Orthovanadate and 1X Protease Inhibitor cocktail (Sigma). Protein concentrations were quantified using Bradford assay and equal amounts of lysates were loaded per well for Western blot analysis.

### Southern blot analysis

Cells were grown on 25 mm plates and infected at an MOI of 5. Cells were harvested at the indicated timepoints, pelleted and resuspended in Southern Lysis Buffer. Cells were proteinase K treated overnight at 37°C, and sheared using 25 G X 5/8 inch 1 mL needle-syringe (BD Biosciences). Total DNA content was quantified using Nanodrop, equal amount of DNA loaded per well and electrophoresed on a 1 percent agarose gel. Samples were transferred to a nitrocellulose membrane and hybridized with completely homologous genomic clones.

### Antibodies

Commercially available antibodies were used for ChIP assays and Immunofluorescence, and are described in the Antibody Table (*Table 2*) and Key Resources Table.

### Plasmids

Lenti-CRISPRv2 plasmid was produced by Feng Zhang (Addgene plasmid 52961, [*Sanjana et al., 2014*]). pgRNA-humanized plasmid was produced by Stanley Qi (Addgene plasmid 44248, [*Qi et al.,*

**Table 2.** Antibody table.

| Antibody | Concentration used |
|---|---|
| NS1 (see Key Resources Table) | ChIP: 6 µg<br>Immunofluorescence:<br>2 µg IB: 2 µg |
| γ-H2AX (anti-mouse); EMD Millipore | ChIP: 5 µg |
| γ-H2AX (anti-rabbit); Abcam | ChIP: 5 µg<br>Immunofluorescence:<br>2 µg IB: 2 µg |
| BRCA1 (anti-mouse); Thermo Fisher Scientific | ChIP: 5 µg |
| FANCD2 (anti-rabbit); Bethyl Laboratories | Immunofluorescence: 2 µg |
| NR5A2 (anti-rabbit); Abcam | Immunofluorescence: 2 µg |
| IgG (mouse); Cell Signaling | ChIP: 5 µg |
| AF 488; anti-rabbit secondary, Life Technologies | Immunofluorescence: 1 µg |
| AF 568; anti-mouse secondary, Life Technologies | Immunofluorescence: 1 µg |
| AF 555; anti-rabbit secondary, Life Technologies | Immunofluorescence: 1 µg |
| AF 647; anti-mouse secondary, Life Technologies | Immunofluorescence: 1 µg |

DOI: https://doi.org/10.7554/eLife.37750.016

*2013*]). pCMV-CD4 was a gift from Dr. Marc Johnson (University of Missouri). Plasmids and reagents are available upon request.

## Quantification and statistical analysis

Imaging studies (3D-FISH and Immunofluorescense) were quantified using ImageJ. Background noise was filtered out using the Kalman Stack Filter plugin, and the 3D distance between viral and cellular genome probes were calculated using Sync Measure 3D plugin to calculate the location of the center of mass between the imaged foci. The 3D-distance was calculated by computing the displacement vector between the two locations. The distances between foci were measured for multiple cells in preparations of viral infections, and were statistically analyzed using GraphPad Prism software. Statistical tests were performed using GraphPad Prism for imaging studies and chromosome conformation capture assays. The relevant statistical tests have been indicated in the respective figure legends. The code for bioinformatics analyses used to process V3C-seq and ChIP-seq data have been tabulated below:

## Data availability

The V3C-seq and ChIP-seq data generated have been deposited in the Gene Expression Omnibus (GEO) under the accession codes GSE112957.

## Acknowledgements

We thank the members of the Pintel Lab for valuable discussions, Richard Adeyemi (Harvard University School of Medicine) for valuable suggestions, Andrew Huber (St. Jude Children's Research Hospital) for helping to purify monoclonal anti-NS1 antibody from hybridoma supernatants, Lisa Burger for expert technical assistance and Alexander Jurkevich (University of Missouri Molecular Cytology Core) for assistance in GSD/dSTORM imaging. GSD/dSTORM imaging was supported by a research award from the University of Missouri Molecular Cytology Core to KM. High-throughput sequencing services were performed at the University of Missouri DNA Core Facility. Computational analysis was performed on the high performance computing infrastructure provided by Research Computing Support Services and in part by the National Science Foundation under grant number CNS-1429294 at the University of Missouri, Columbia MO. Work in the Pintel Laboratory is supported by NIH grants AI046458 and AI116595 to DJP. KM is supported by a Ruth L Kirschstein Postdoctoral Individual National Research Service Award AI131468.

## Additional information

### Competing interests
Matthew S Fuller: is employed by Ultragenyx Pharmaceutical. There are no other competing interests to declare. The other authors declare that no competing interests exist.

### Funding

| Funder | Grant reference number | Author |
|---|---|---|
| National Institute of Allergy and Infectious Diseases | AI046458 | David J Pintel |
| National Institute of Allergy and Infectious Diseases | AI131468 | Kinjal Majumder |
| National Institute of Allergy and Infectious Diseases | AI116595 | David J Pintel |

The funders had no role in study design, data collection and interpretation, or the decision to submit the work for publication.

### Author contributions
Kinjal Majumder, Conceptualization, Data curation, Formal analysis, Funding acquisition, Validation, Investigation, Visualization, Methodology, Writing—original draft, Writing—review and editing; Juexin Wang, Data curation, Software, Formal analysis; Maria Boftsi, Formal analysis, Investigation; Matthew S Fuller, Conceptualization, Investigation, Methodology; Jordan E Rede, Investigation; Trupti Joshi, Software, Formal analysis; David J Pintel, Conceptualization, Supervision, Funding acquisition, Writing—original draft, Project administration, Writing—review and editing

### Author ORCIDs
David J Pintel http://orcid.org/0000-0002-9959-2848

### Decision letter and Author response
Decision letter https://doi.org/10.7554/eLife.37750.028
Author response https://doi.org/10.7554/eLife.37750.029

## Additional files

### Supplementary files
• Supplementary file 1. contains a table with the sequences of PCR primers and Taqman probes used in this study.
DOI: https://doi.org/10.7554/eLife.37750.017
• Transparent reporting form
DOI: https://doi.org/10.7554/eLife.37750.018

### Data availability
Sequencing data have been deposited in GEO under accession code GSE112957.

The following dataset was generated:

| Author(s) | Year | Dataset title | Dataset URL | Database, license, and accessibility information |
|---|---|---|---|---|
| Kinjal Majumder, Juexin Wang, Maria Boftsi, Matthew S Fuller, Jordan E Rede, Trupti Joshi, David J Pintel | 2018 | Parvovirus Minute Virus of Mice Localizes to Sites of Cellular DNA Damage to Establish and Amplify its Lytic Infection | http://www.ncbi.nlm.nih.gov/geo/query/acc.cgi?acc=GSE112957 | Publicly available at the NCBI Gene Expression Omnibus (accession no: GSE112957) |

The following previously published datasets were used:

| Author(s) | Year | Dataset title | Dataset URL | Database, license, and accessibility information |
|---|---|---|---|---|
| Rao SS, Huntley MH, Durand NC, Stamenova EK | 2014 | A three-dimensional map of the human genome at kilobase resolution reveals prinicples of chromatin looping | https://www.ncbi.nlm.nih.gov/geo/query/acc.cgi?acc=GSE63525 | Publicly available at the NCBI Gene Expression Omnibus (accession no: GSE63525) |
| Kraushaar DC, Jin W, Maunakea A, Abraham B | 2013 | Genome-wide incorporation dynamics reveal distinct categories of turnover for the histone variant H3.3 | https://www.ncbi.nlm.nih.gov/geo/query/acc.cgi?acc=GSE51505 | Publicly available at the NCBI Gene Expression Omnibus (accession no: GSE51505) |
| Barlow JH, Faryabi RB, Callén E, Wong N | 2013 | Genome-wide mapping of early replication fragile sites (ERFs) | https://www.ncbi.nlm.nih.gov/geo/query/acc.cgi?acc=GSE43504 | Publicly available at the NCBI Gene Expression Omnibus (accession no: GSE43504) |

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
