## [Decision Letter]

Thank you for submitting your article "Parvovirus Minute Virus of Mice Interacts with Sites of Cellular DNA Damage to Establish and Amplify its Lytic Infection" for consideration by *eLife*. Your article has been reviewed by three peer reviewers, one of whom is a member of our Board of Reviewing Editors, and the evaluation has been overseen by Jessica Tyler as the Senior Editor. The following individual involved in review of your submission has agreed to reveal her identity: Alison Anne McBride (Reviewer #2).

The reviewers have discussed the reviews with one another and the Reviewing Editor has drafted this decision to help you prepare a revised submission.

The manuscript presents a novel, high-throughput viral chromosome capture assay for use in trans (termed V3C-Seq) to enable a genome-wide analysis of direct interactions between DNA viruses and cellular chromatin. This manuscript nicely demonstrates that Virus Association Domains correlate strongly with sites of DNA damage during MVM infection. Furthermore, V3C-Seq should be useful in characterizing viral-host interactions for other DNA viruses, making this study a valuable contribution to the virology field.

We ask you to address the review points in the revision, especially distinguishing your technique from others, the resolution of the VAD mapping and statistical analysis of your data, discussing your detection of genomes from the large replicating pool, and that you are mapping replicating genomes, not the targeting of input genomes.

*Reviewer #1:*

I have a few points that could improve the manuscript.

1) The VADs are fairly large as defined, ranging from 5-15 megabases. Thus, it is hard to see if the VADs really map to the same "site" as DNA damage. Some higher resolution presentation could help to clarify this.

2) The authors state that their method is the first to map "trans" interactions but cite Jiang et al., 2017 as having done this with ChIA-PET. It would be helpful to distinguish the two techniques better.

3) There is a very limited discussion of the literature on how localization to DNA damage sites helps replication of the virus.

4) Do the MVM genomes target to sites of the DNA damage repair apparatus or to cellular DNA sites?

5) The authors should use "discrete" uniformly throughout the manuscript.

6) Results. The overlap of sites is described as significant, but some statistical analysis should be provided to document this.

Reviewer #2:

The authors present a novel, high-throughput viral chromosome conformation capture assay for use in trans (termed V3C-seq) to enable a genome-wide analysis of direct interactions between DNA viruses with cellular chromatin. This article nicely demonstrates that Virus Association Domains (VADs) correlate strongly with sites of DNA damage, during MVM infection. Furthermore, V3C-seq should be useful for characterizing viral-host interactions for other DNA viruses, making this study a valuable contribution to the virology field.

• In the Introduction, the authors state that an unbiased way to map sites of viral-host interaction has been largely unavailable until recently. However, in situ Hi-C has previously been used to map interactions between EBV, KSHV and HPV and host chromosomes (Moquin et al., 2018). The authors should emphasize why the V3C-seq method is novel, i.e. in terms of inverse PCR to yield V3C-seq DNA libraries.

• Figure 1A should include controls of NR5A2 immunostaining and immunostaining of mock-infected A9 cells to show normal levels of FANCD2 and γ-H2AX in these cells, and as a negative control for any non-specific NS1 signal. 1B is very confusing: should the Y axis read "Distance from NS1 APAR bodies? It is not clear what "foci" are being referred to.

• Subsection “The replicating MVM genome localized adjacent to regions of the cellular genome undergoing a DDR”. Although gH2AX does characteristically amplify on megabases of cellular DNA, I do not think it is accurate to say that this "precludes the possibility" of it marking the MVM genome.

• Figure 2: a more detailed map of the viral genome with HindIII, NlaIII, nucleotide numbers and P4 and P38 promoters would be helpful.

• Figure 3D: set the heatmap scales to the same value.

• Figure 5: Panel B, include inset for higher magnification image of localization of MVM with VAD as shown for Figure 1D. It is very difficult to see as is. Panel C, please indicate the alternative 6pA allele. Scale bars should be added. The signal for MVM FISH seems surprisingly small for 16 hours p.i. (especially compared to images of NS1 at this time point in Figure 1). There appears to be localized signal for MVM and 6pA at the nuclear border making it difficult to interpret the result. Please include additional representative image(s).

• Do VADs overlap with common fragile sites, as well as the ERFs? These are often transcriptionally active, encode long genes and are late replicating in the cell cycle.

• Authors' state in the Discussion: "In this study we have utilized chromosome conformation capture technology for the first time to map the trans interaction of a lytic virus with the cellular genome in a non-biased way". Again, Moquin et al., 2018, use Hi-C to map EBV to cellular chromosomes during latent and lytic phases of the life cycle. This statement should be addressed accordingly to emphasize the novel aspects of V3C-seq relative to Hi-C for addressing viral-host interactions.

Reviewer #3:

In this very comprehensive and rather jargon-dense paper, the authors have adapted a chromosome conformation capture (3C) technique previously used for exploring interactions between cis-related DNA sequences in chromatin to a trans configuration, in order to explore the association of intracellular forms of parvoviral DNA with host chromosomal sites undergoing DNA damage response (DDR)-initiated repair.

The description of the V3C-seq procedure given in Figure 2 and the text is quite clear in principal, and should be useable by others wishing to apply this approach in their own virus:host system. However, the details given for MVM are somewhat confusing. There are 2 HindIII sites in the MVMp genome, of which the inverse primers used target the one at nt 2651. Secondary digestion with NlaIII, which cuts at 33 sites would leave a ~750 bp MVM fragment attached to the associated cellular DNA. This fragment lies between nts 1899 and 2651, and covers the P38 promoter – it is not clear, then, why the authors refer to this as containing the P4-P38 4C inverse primers – what does this mean? How does this fragment relate to the P4 promoter? A clearer Figure 2 explaining this in detail would be helpful.

The authors present a number of experiments employing different independent methodologies to corroborate their findings with V3C-seq, that intranuclear MVM duplex genomes associate with regions that have sustained DNA damage that may, or may not, be caused by the infection. These are important additional experiments, since they help to rule out the possibility that V3C merely identifies association between MVM and host DNA because cross-linking can only occur at sites of DNA damage, perhaps as a result of local opening up of the chromatin.

There is one aspect of the discussion ongoing throughout the paper that is of concern, and that is the repeated reference to how the results indicate that incoming MVM genomes are targeted to sites of DNA damage. This important aspect of infection initiation is not addressed by either the techniques used or the timing of the experiments presented. At all the times post-release examined in this paper, there may be hundreds, if not thousands, of duplex MVM genomes present in each APAR body. Each of these molecules presents one or more terminal structure that would almost certainly be detected as a double-strand break by host DDR systems. Additionally, replication proceeds by strand-displacement, yielding hundreds to thousands of nucleotides of single-stranded DNA attached to each replicative intermediate, leading to extensive RPA2-dependent signaling. The presence of these forms of viral DNA in each developing APAR body makes it extremely difficult to ascertain the extent to which the DDR signals emanating from them are due to damage to cellular DNA. This aspect of the overall DDR observed in infected cell nuclei is not discussed.

Finally, one hopes that the software used to analyze the data generated in this paper works better than that used to generate the bibliography, which is a mess!

"initial interaction points" – the results presented do not address where input MVM genomes initially localize (see general comments).

Subsection “The MVM genome associated directly with discrete sites on the cellular genome”, third paragraph: while inhibition of packaging allows further accumulation of viral RF DNA molecules, packaging itself would not reduce the abundance of viral RF DNA, since the single strands destined for packaging are generated from existing RF molecules by strand-displacement synthesis.

"which covalently binds to the viral genome" is a confusing statement. As a consequence of replication there is one NS1 molecule covalently bound to the 5' end of each viral DNA strand, whereas there are multiple sites throughout the viral genome to which NS1 can bind non-covalently, an observation that may be highly relevant to the identity and functionality of APAR bodies.

"MVM first localized" – see concern in the second paragraph of the subsection “The MVM genome associated directly with discrete sites on the cellular genome”.

"supported by the present work" is an over reach – the current paper does not address where the incoming genomes locate.

---

## [Author Response]

Reviewer #1:I have a few points that could improve the manuscript.1) The VADs are fairly large as defined, ranging from 5-15 megabases. Thus, it is hard to see if the VADs really map to the same "site" as DNA damage. Some higher resolution presentation could help to clarify this.

Although high resolution snapshots were originally provided in Figure 3—figure supplement 2C (in the 600-750 kB range), we have added additional examples and at even higher resolution at selected regions of the larger VADs demarcated by red rectangles (85-170 kB range, also in Figure 3—figure supplement 2C, right panel). As described (subsection “The MVM genome initiate infection at sites of cellular DNA damage that in mock infected cells also exhibited DNA damage as the cells cycled through S-phase, and as infection progressed, localized to additional sites of induced damage”, third paragraph), these clearly show VADs and “sites” of DNA damage, as evidenced by γ-H2AX ChIP-seq clearly overlapping.

2) The authors state that their method is the first to map "trans" interactions but cite Jiang et al., 2017 as having done this with ChIA-PET. It would be helpful to distinguish the two techniques better.

The distinction between our 4C analyses and ChIA-PET have been discussed and clarified in the manuscript Discussion (fourth paragraph).

3) There is a very limited discussion of the literature on how localization to DNA damage sites helps replication of the virus.

As this is the first report to address localization of a parvovirus genome to sites of cellular DNA damage, we assume the reviewer is asking about how localization to DNA damage sites might help replication of DNA virus genomes in general. This is an important point. There is significant precedent for DNA pol-δ (which is the replicative enzyme for MVM), RNA pol II, and numerous RNA processing enzymes localizing to sites of DNA damage, which would provide a rich environment for viral replication. We expand on the literature background for this knowledge in the appropriate sections of the Introduction (fourth paragraph) and Discussion (first paragraph), and add the appropriate references, to support our claims to this effect.

4) Do the MVM genomes target to sites of the DNA damage repair apparatus or to cellular DNA sites?

It is not yet clear whether the MVM genomes directly target damaged DNA or the repair apparatus closely associated with these sites. We plan to use ChIP-re-ChIP type experiments to probe this question.

5) The authors should use "discrete" uniformly throughout the manuscript.

We have made the use of “discrete” uniform, and occasionally used changed the wording to “distinct”. Also suggested by reviewer #3.

6) Results. The overlap of sites is described as significant, but some statistical analysis should be provided to document this.

We have performed Jaccard analysis to quantify the statistical significance of the overlap, and have presented them in Figure 2C, right panels (V3C-seq Peak Correlations).

Reviewer #2:[…] • In the Introduction, the authors state that an unbiased way to map sites of viral-host interaction has been largely unavailable until recently. However, in situ Hi-C has previously been used to map interactions between EBV, KSHV and HPV and host chromosomes (Moquin et al., 2018). The authors should emphasize why the V3C-seq method is novel, i.e. in terms of inverse PCR to yield V3C-seq DNA libraries.

We have modified the Introduction (last paragraph) and Discussion (fourth paragraph) to explain these differences and the value of using our technique.

• Figure 1A should include controls of NR5A2 immunostaining and immunostaining of mock-infected A9 cells to show normal levels of FANCD2 and γ-H2AX in these cells, and as a negative control for any non-specific NS1 signal. 1B is very confusing: should the Y axis read "Distance from NS1 APAR bodies? It is not clear what "foci" are being referred to.

We have added the control stainings for Figure 1A as suggested; they show NR5A2, FANCD2 and γ-H2AX background levels, as well as the absence of NS1 staining in mock infected cells as expected. Figure 1B has been re-labeled to the now-correct, and more clearly informative, “Distance from NS1 APAR bodies”.

• Subsection “The replicating MVM genome localized adjacent to regions of the cellular genome undergoing a DDR”. Although gH2AX does characteristically amplify on megabases of cellular DNA, I do not think it is accurate to say that this "precludes the possibility" of it marking the MVM genome.

Yes, we agree this was an overstatement. We have modified this claim. Please see subsection “The replicating MVM genome localized adjacent to regions of the cellular genome undergoing a DDR”.

• Figure 2: a more detailed map of the viral genome with HindIII, NlaIII, nucleotide numbers and P4 and P38 promoters would be helpful.

A more detailed map of the MVM genome with HindIII sites, the relevant NlaIII sites, the locations of the inverse PCR primers, and the numerical locations of the restriction sites have been provided at the bottom of 2A. This was also requested by reviewer #3.

• Figure 3D: set the heatmap scales to the same value.

These changes have been made.

• Figure 5: Panel B, include inset for higher magnification image of localization of MVM with VAD as shown for Figure 1D. It is very difficult to see as is. Panel C, please indicate the alternative 6pA allele. Scale bars should be added. The signal for MVM FISH seems surprisingly small for 16 hours p.i. (especially compared to images of NS1 at this time point in Figure 1). There appears to be localized signal for MVM and 6pA at the nuclear border making it difficult to interpret the result. Please include additional representative image(s).

The requested modifications to Figure 5B have been made. We have replaced Figure 5C with a better representative image. We have additionally provided a second representative image of MVM-VAD association (Figure 5B) and MVM-nonVAD association (Figure 5C). We have also provided representative images of the MVM genome (red) with both 19qA VAD (green) and 9pA non-VAD (cyan) in the same nucleus (Figure 5D). These representative images further support our findings of MVM interaction sites at 16hpi using V3C-seq. The observation of the MVM signal in super-resolution microscopy at the nuclear border was likely due to the presence of incoming viral genomes. These observations are being investigated further.

• Do VADs overlap with common fragile sites, as well as the ERFs? These are often transcriptionally active, encode long genes and are late replicating in the cell cycle.

ERFs have been systematically identified in the mouse genome, whereas the CFSs have been primarily identified on the human genome. Focused V3C-qPCR assays have determined that MVM associates with a subset of CFSs on the human genome in NB-324K cells (Figure 4—figure supplement 1C). We have not yet determined the transcriptional and replication status of these sites during MVM infection. However, this is an interesting question and such work will be pursued in the future.

• Authors' state in the Discussion: "In this study we have utilized chromosome conformation capture technology for the first time to map the trans interaction of a lytic virus with the cellular genome in a non-biased way". Again, Moquin et al., 2018, use Hi-C to map EBV to cellular chromosomes during latent and lytic phases of the life cycle. This statement should be addressed accordingly to emphasize the novel aspects of V3C-seq relative to Hi-C for addressing viral-host interactions.

As discussed above, our technique is different because is enables us to interrogate the interactome of a specific region of interest (the MVM genome in this study) in finer detail by utilizing inverse PCRs to amplify the hybrid DNA fragments. While generally similar information can be obtained using Hi-C analyses, a significantly higher level of sequencing depth would be required to reach the same level of resolution because Hi-C assays measure all of the hybrid junctions formed by the chromosome conformation capture technique (also making it significantly more cost prohibitive). Utilization of V3C-seq allows the mapping of these interactomes in replicates over timecourses of infection and provides finer, more detailed interaction data in a more cost-efficient way. As described above, we will modify the Discussion (fourth paragraph) and Introduction (last paragraph) to more clearly convey these differences.

Reviewer #3:[…] The description of the V3C-seq procedure given in Figure 2 and the text is quite clear in principal, and should be useable by others wishing to apply this approach in their own virus:host system. However, the details given for MVM are somewhat confusing. There are 2 HindIII sites in the MVMp genome, of which the inverse primers used target the one at nt 2651. Secondary digestion with NlaIII, which cuts at 33 sites would leave a ~750 bp MVM fragment attached to the associated cellular DNA. This fragment lies between nts 1899 and 2651, and covers the P38 promoter – it is not clear, then, why the authors refer to this as containing the P4-P38 4C inverse primers – what does this mean? How does this fragment relate to the P4 promoter? A clearer Figure 2 explaining this in detail would be helpful.

We apologize for the confusion. Chromosome conformation capture assays quantify the formation of hybrid DNA junctions that are generated between distally located DNA fragments based only upon the primary restriction enzyme used (which, in these assays, is HindIII). Any crosslinks on the 5’ end of the MVM genome between the viral left-hand end (LHE) and the viral P38 (capsid gene) promoter will enable the formation of a hybrid junction at the HindIII site at 2651. Upon formation of these libraries of HindIII junctions, NlaIII is utilized as a secondary restriction enzyme to generate a smaller DNA library for the purpose of high-throughput sequencing. We have clarified this point in the text (subsection “The MVM genome associated directly with discrete sites on the cellular genome”, second paragraph) and have provided a more detailed version of the MVM genome at the bottom of Figure 2A incorporating the details of the MVM genome. This should allow a clearer understanding of our protocol.

The authors present a number of experiments employing different independent methodologies to corroborate their findings with V3C-seq, that intranuclear MVM duplex genomes associate with regions that have sustained DNA damage that may, or may not, be caused by the infection. These are important additional experiments, since they help to rule out the possibility that V3C merely identifies association between MVM and host DNA because cross-linking can only occur at sites of DNA damage, perhaps as a result of local opening up of the chromatin.

Yes, this is why we performed these experiments.

There is one aspect of the discussion ongoing throughout the paper that is of concern, and that is the repeated reference to how the results indicate that incoming MVM genomes are targeted to sites of DNA damage. This important aspect of infection initiation is not addressed by either the techniques used or the timing of the experiments presented. At all the times post-release examined in this paper, there may be hundreds, if not thousands, of duplex MVM genomes present in each APAR body. Each of these molecules presents one or more terminal structure that would almost certainly be detected as a double-strand break by host DDR systems. Additionally, replication proceeds by strand-displacement, yielding hundreds to thousands of nucleotides of single-stranded DNA attached to each replicative intermediate, leading to extensive RPA2-dependent signaling. The presence of these forms of viral DNA in each developing APAR body makes it extremely difficult to ascertain the extent to which the DDR signals emanating from them are due to damage to cellular DNA. This aspect of the overall DDR observed in infected cell nuclei is not discussed.

There are two main points here of concern to the reviewer.

The first concern relates to targeting of the incoming virus genome. As the incoming parvovirus genome is single-stranded it is not amenable to V3C-seq. Thus we chose to first examine the earliest time post-infection when the double stranded replicative form was sufficient to validate our technique. We agree that there are likely many copies of these viral replicative forms even at these early times; however, as mentioned in the fourth paragraph of the Results subsection “The MVM genome initiate infection at sites of cellular DNA damage that in mock infected cells also exhibited DNA damage as the cells cycled through S-phase, and as infection progressed, localized to additional sites of induced damage”, third paragraph”, we find a strong association of MVM with sites that in uninfected cells exhibit DNA damage upon replication. We proposed that this suggested that MVM may initially establish replication at cellular fragile sites that are susceptible to DNA damage as cell cycle through S-phase. We did not mean to imply that we had unequivocally proven that incoming genomes themselves necessarily migrate directly to these specific sites. We will discuss the reviewer’s point about abundant genomes (Discussion, first paragraph), and we will temper our conclusion to say that our results are “consistent” with this possibility (subsection “The MVM genome initiate infection at sites of cellular DNA damage that in mock infected cells also exhibited DNA damage as the cells cycled through S-phase, and as infection progressed, localized to additional sites of induced damage”, third paragraph”, fourth and last paragraphs; Discussion, first paragraph), and that specific localization of incoming genomes remains circumstantial (Discussion, first paragraph). Additionally, we will change “initially established” to “established” in the Abstract. This is really a matter of sensitivity, currently under further characterization, and we agree that this should be presented more clearly.

The reviewer’s second point is related to the first, and certainly was a target of our attention. As the reviewer points out, because there are many replicating viral genomes at sites of their interaction with the cellular chromosome, it is difficult to distinguish DDR signaling due to virus replication from that due to damage on cellular DNA. However, until we can unequivocally address this point using either gradient separation techniques or native ChIPs to effectively isolate viral chromosomes for analysis, we have presented an experiment that strongly suggests that the damage we see is at least primarily due to damage of the cellular chromosome. As we point out in the last paragraph of the Results subsection “VM infection induced distinct sites of cellular DNA damage as demonstrated by ChIP-seq for γ-H2AX”, in addition to the association of MVM with sites that in uninfected cells exhibit DNA damage upon progression into S-phase, VADs also correlated strongly with γ-H2AX ChIP-seq sites identified on cellular chromosomes of non-infected, HU-treated, A9 cells (Figure 3A). This implies that the γ-H2AX identified associated with MVM during infection resides on cellular DNA; however, we will add the phrase “but does not prove” to our conclusion in the Results subsection “The MVM genome initiate infection at sites of cellular DNA damage that in mock infected cells also exhibited DNA damage as the cells cycled through S-phase, and as infection progressed, localized to additional sites of induced damage”, fourth paragraph”. This point is treated as non-conclusive in the Discussion (second paragraph.)

Finally, one hopes that the software used to analyze the data generated in this paper works better than that used to generate the bibliography, which is a mess!

The bibliography errors have been corrected. Sorry.

"initial interaction points" – the results presented do not address where input MVM genomes initially localize (see general comments).

This is discussed in detail above. We will change this phrase to “early amplification sites”.

Subsection “The MVM genome associated directly with discrete sites on the cellular genome”, third paragraph: while inhibition of packaging allows further accumulation of viral RF DNA molecules, packaging itself would not reduce the abundance of viral RF DNA, since the single strands destined for packaging are generated from existing RF molecules by strand-displacement synthesis.

This is likely correct. This was a minor point, and we will delete this. This in no way affects our model.

"which covalently binds to the viral genome" is a confusing statement. As a consequence of replication there is one NS1 molecule covalently bound to the 5' end of each viral DNA strand, whereas there are multiple sites throughout the viral genome to which NS1 can bind non-covalently, an observation that may be highly relevant to the identity and functionality of APAR bodies.

We have clarified this better in the last paragraph of the subsection “The MVM genome initiate infection at sites of cellular DNA damage that in mock infected cells also exhibited DNA damage as the cells cycled through S-phase, and as infection progressed, localized to additional sites of induced damage”.

"MVM first localized" – see concern in the second paragraph of the subsection “The MVM genome associated directly with discrete sites on the cellular genome”.

We agree fully and will change this as explained and indicated in detail above.

"supported by the present work" is an over reach – the current paper does not address where the incoming genomes locate.

This is the same point as above. We agree fully and address this in detail in the text as described above.